Manuscript prepared for Atmos. Chem. Phys.
with version 2015/04/24 7.83 Copernicus papers of the LaTeX class ./latex/copernicus.cls.
Date: 24 May 2016

# Diurnal variation of tropospheric relative humidity in tropical region

Isaac Moradi[1,2], Philip Arkin[1], Ralph Ferraro[2], Patrick Eriksson[3], and Eric Fetzer[4]

[1]ESSIC, University of Maryland, College Park, Maryland, USA.
[2]STAR, NOAA, College Park, Maryland, USA.
[3]Chalmers University of Technology, Gothenburg, Sweden.
[4]Jet Propulsion Laboratory (JPL), CalTech, California, USA.

*Correspondence to:* Isaac Moradi, NASA's Goddard Space Flight Center, Code 610.1, Greenbelt, MD 20771. (isaac.moradi@nasa.gov)

**Abstract.** Despite the importance of water vapor especially in the tropical region, the diurnal variations of water vapor have not been completely investigated in the past due to the lack of adequate observations. Measurements from Sondeur Atmosphérique du Profil d'Humidité Intertropicale par Radiométrie (SAPHIR) onboard the low inclination Megha-Tropiques satellite with frequent daily revisits provide a valuable dataset for investigating the diurnal and spatial variation of tropospheric relative humidity in the tropical region. In this study, we first transformed SAPHIR observations into layer-averaged relative humidity, then partitioned the data based on local observation time into 24 bins with a grid resolution of one degree. Afterwards, we fitted Fourier series to the binned data. Finally, the mean, amplitude, and diurnal peak time of relative humidity in tropical region were calculated for each grid point using either the measurements or Fourier series. The results were separately investigated for different SAPHIR channels as well as for relative humidity with respect to both liquid and ice phases. The results showed that the wet and dry regions are, respectively, associated with convective and subsidence regions which is consistent with the previous studies. The mean tropospheric humidity values reported in this study are generally 10% to 15% higher than those reported using infrared observations which is because of strict cloud screening for infrared measurements. The results showed a large inhomogeneity in diurnal variation of tropospheric relative humidity in tropical region. The diurnal amplitude was larger over land than over ocean and the oceanic amplitude was larger over convective regions than over subsidence regions. The results showed that the diurnal amplitude is less than $10\,\%$ in middle and upper troposphere, but it is up to $30\,\%$ in lower troposphere over land. Although, the peak of RH generally occurs over night or in early morning, there are several regions where the diurnal peak occurs at other times of the day. The early morning peak time is because of a peak in convective activities in early morning. Additionally, a double peak was observed in tropospheric humidity over some regions which is consistent with double peak in precipitation.

## 1 Introduction

Water vapor is the dominant natural greenhouse gas in the atmosphere, thus it significantly influences the Earth's climate and energy budget. Water vapor is responsible for nearly half of the poleward and most of the upward heat transfer, and also affects the Earth's hydrologic cycle through evaporation and condensation (Sherwood et al., 2010). In addition, water vapor drives extreme weathers such

as rainstorm, floods, and the initiation of convective cyclones (Keil et al., 2008). Water vapor in the free troposphere, the layer expanding from 1-2 km above the surface up to tropopause, strongly contributes to the water vapor feedback through radiative processes (Trenberth et al., 2009; Dessler and Sherwood, 2009), with maximum feedback occurring in the tropical free troposphere (Dessler et al., 2008). Therefore, water vapor is expected to also play an important role in global warming and

climate change predictions (Cess et al., 1990). For instance, assuming a constant relative humidity (RH) in climate models doubles the rise in temperature compared to when the water vapor feedback is forced to be zero (Minschwaner and Dessler, 2004; Soden et al., 2005). Additionally, the net cooling of the atmosphere is indirectly affected by tropospheric water vapor through the initiation of clouds and convective heating (Sherwood, 2010). Diurnal cycles in temperature and moisture drive

diurnal variations in temperature, precipitation, and convective activities (Chung et al., 2007), therefore, are expected to interact significantly with, for example, changes in global mean humidity or temperature. However, current climate and numerical weather prediction models do not adequately simulate the diurnal variation of tropospheric humidity (Chung et al., 2013), a failing that is very likely to lead to inaccuracies in their simulations. As models are improved, accurate observations of

diurnal cycles of humidity will be crucial in verifying the validity of simulations.

Most studies in the past have mainly used infrared (IR) satellite data from geostationary orbits to evaluate the diurnal cycle of RH (e.g., Soden, 2000; Tian, 2004; Chung et al., 2007). Chung et al. (2013) evaluated the diurnal variation of Upper Tropospheric Humidity (UTH) in reanalysis using Meteosat-5 data and reported a distinct diurnal cycle of UTH over tropical convective regions.

Schröder et al. (2014) developed a homogenized dataset for free tropospheric humidity under clear sky conditions. Although, this dataset is valuable in clear-sky conditions especially over land, however it is biased towards dry conditions in the tropical region as most of the IR measurements are filtered out over the subsidence regions because of the presence of deep-convective clouds. Chung et al. (2007) evaluated the diurnal variations of RH over Africa using IR observations. They reported

a large diurnal amplitude over land but a smaller amplitude over oceanic subsidence regions. In an early attempt to relate spatial distribution of tropospheric humidity with physical mechanisms, Udelhofen and Hartmann (1995) evaluated the decrease in relative humidity with distance from the edge of the clouds. They reported that the rate of decrease in UTH is lower in Intertropical Convergence Zone (ITCZ) than in the subsidence regions showing a wider impact for the ITCZ clouds

on the environment surrounding the clouds. It should be noted that IR observations are very sensitive to clouds, thus the data need to be strictly filtered for clouds before being analyzed. The cloud

screening removes a large portion of the IR measurements especially over convective regions. The rejected observations normally represent moist conditions, therefore the IR results only represent dry conditions. For instance, John et al. (2011) indicated that the IR cloud screening introduces on average around 10 % systematic error in the upper tropospheric humidity values. It is clearly shown in John et al. (2011) that the cloud screening especially removes most of the data over the convective regions causing a large systematic bias in the RH analysis for the convective regions. It should be noted that among the channels, the upper tropospheric channels are less sensitive to clouds than the lower channels, because the weighting functions for the upper channels normally peak above the clouds, therefore it is expected that the dry bias due to cloud screening is even larger for the middle and lower tropospheric channels. It should be noted that the cloud screening not only impacts the RH amplitude by removing the moist conditions, but it also impacts the diurnal peak time. Thus, the analyses performed using IR observations are biased for both the spatial distribution and the diurnal cycle of the RH values.

One exception is Kottayil et al. (2013) that used multi-instrument microwave measurements from five polar-orbiting satellites to investigate the diurnal variation of brightness temperature (Tb) over the globe. However, other issues are involved when data from polar-orbiting instruments are utilized. First, polar-orbiting satellites only overpass each location twice a day, thus even a constellation of five satellites do not properly represent the diurnal variation of RH (e.g. see Figure 1 in Kottayil et al. (2013) for the temporal coverage in different years). The orbital drift only slightly enhances the temporal coverage of the data. Second, the multi-instrument differences are an important issue when data are combined. Kottayil et al. (2013) used a dataset that was inter-calibrated using Simultaneous Nadir Observations (SNO). The inter-satellite differences are normally scene dependent, however SNO's normally happen in the polar region so cannot sufficiently resolve the scene dependency due to non-linearity in the calibration of microwave instruments. The observations discussed above are all downward looking, covering altitudes up to about 300 hPa. Diurnal RH variations at higher altitudes have been studied by microwave limb sounding data. Coarse estimates, having a resolution of 6-hour in local time, were provided by Eriksson et al. (2010), by combining data from two different sun-synchronous satellites, Aura Microwave Limb Sounder (MLS) and Odin Sub-Millimeter Radiometer (SMR). These estimates were compared to some climate models and it was found that the models tend to underestimate diurnal variations and partly also simulate maximum RH at wrong local time. Later, Eriksson et al. (2014) derived diurnal variations using the Superconducting Submillimeter-Wave Limb-Emission Sounder (SMILES) instrument. SMILES measurements are only available for a 6-month period, but the measurements are made at different local times so suitable for evaluating the diurnal variations. The SMILES data were found to confirm the main features reported by Eriksson et al. (2010).

In summary, despite the importance of water vapor and its distribution in time and space, there still exists a considerable uncertainty in our knowledge of the diurnal and spatial distribution of tro-

pospheric water vapor. The efforts so far have not been able to clearly determine the temporal and spatial distribution of water vapor in the atmosphere due to the lack of adequate observations. This study benefits from observations from Sondeur Atmosphérique du Profil d'Humidité Intertropicale par Radiométrie (SAPHIR) onboard Megha-Tropiques (M-T), a low-inclination satellite with frequent revisits in tropical region between 35 °N and 35 °S. SAPHIR is equipped with six water vapor channels sensitive to upper to lower tropospheric RH. SAPHIR provides a great opportunity to analyze the diurnal and spatial variation of RH in the tropical region using data from a single instrument (Moradi et al., 2015b). As stated before, relative humidity is the ratio of water vapor pressure to the saturated vapor pressure. The water vapor pressure depends mainly on the water vapor content of the atmosphere, but the saturated vapor pressure depends on the air temperature. Therefore, the diurnal variation of RH does not necessary indicate change in the water vapor content of the atmosphere, because it is affected by both diurnal variation of water vapor and air temperature. For instance, change in the amount of lower tropospheric water vapor over deserts is very small during day, but RH can significantly change because of change in air temperature. Therefore, it is more desired to analyze the diurnal variation of absolute humidity parameters. However, measurements from microwave water vapor channels are most sensitive to change in RH and cannot be used to derive absolute humidity parameters. The rest of the paper is organized as follows: Section 2 discusses satellite data used in this study, Section 3 presents the methodology including the satellite Tb to RH transformation method as well as Fourier series, Section 4 discusses the results, and Section 5 summarizes the study.

## 2   Satellite Data

Megha-Tropiques is a low-inclination satellite launched in November 2011 that frequently visits the tropical band between 35 °S and 35 °N. SAPHIR is a microwave humidity sounder onboard the M-T satellite that measures tropospheric RH using six channels centered around the water vapor absorption line at 183 GHz. All SAPHIR channels have double pass-band with horizontal polarization operating at $183 \pm 0.20$, $183 \pm 1.10$, $183 \pm 2.80$, $183 \pm 4.20$, $183 \pm 6.80$, and $183 \pm 11.0$ GHz. The instrument swath width is 1700 km, and the resolution is 10 km at nadir for all the channels. Figure 1 shows the weighting functions for the SAPHIR channels which are roughly sensitive to upper (channel 1 peaking around 10 km) to lower troposphere (channel 6 peaking around 2 km). It should be noted that one limitation of the measurements from microwave (as well as infrared) humidity sounders is that the peak of Jacobians changes with the water vapor content of the atmosphere. For instance shaded regions in Figure 1 depict the range of Jacobinas for SAPHIR channels derived from the European Organization for the Exploitation of Meteorological Satellites (EUMETSAT) database (Chevallier et al., 2006). However, this shift is expected to be small in tropical region and should

not affect the results. We used SAPHIR L1A data, for the period January 2012 to September 2015, processed by the Centre National d'Etudes Spatiales (CNES).

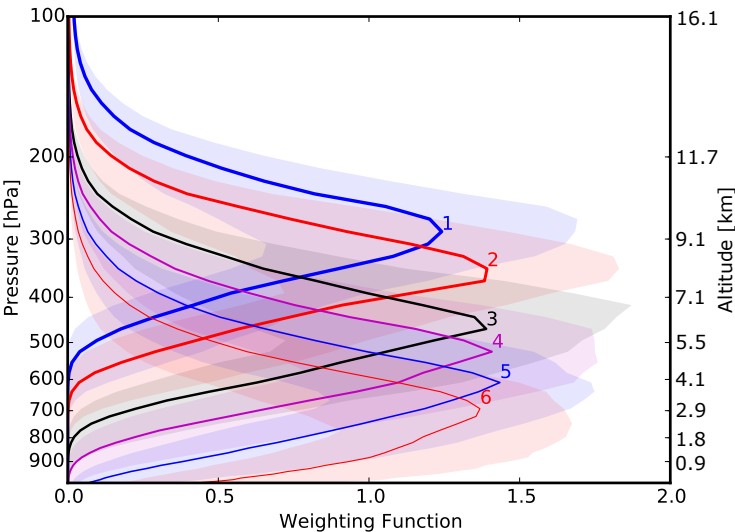

**Figure 1.** The weighting functions for the SAPHIR channels calculated using a subset of EUMETSAT profiles. The selected subset includes 5000 profiles, so that the shaded areas show the range for 25th and 75th percentiles of all profiles. The solid lines show the 50th percentile and the channels' numbers are printed on the plot.

## 3  Methodology

This section discusses the methodology that is used to transform satellite radiances into RH and also Fourier series that are used to investigate the diurnal cycle of tropospheric RH.

### 3.1  Satellite Tb to RH Transformation

A simple method that was developed by Soden and Bretherton (1993) has been widely used in the past to convert satellite microwave measurements to layer averaged RH (e.g., Buehler and John, 2005; Moradi et al., 2010, 2015a). In this simple relation, the satellite Tb's are linearly related to the natural logarithm of layer averaged humidity as follows:

$$ln(RH^{ch}) = a^{ch} + b^{ch} \times Tb^{ch} \tag{1}$$

where $a$ and $b$ are empirical coefficients that change with the earth incidence angle unless the satellite Tb's are corrected for the limb-effect, and the ch stands for the channels. Since calculating the empirical coefficients as a function of earth incidence angle introduces a very large look-up table; similar to Moradi et al. (2015a), we first applied a limb-correction technique to SAPHIR Tb's then used the same empirical coefficients for all the incidence angles. We calculated the limb-darkening

($\Delta Tb$) as the difference between Tb for each beam position and corresponding nadir Tb using data averaged over a long-period of time:

$$Tb^n = Tb(\theta) - \Delta Tb$$

$$\Delta Tb = c \times \ln(cos\theta) \tag{2}$$

where $Tb^n$ is Tb at sub-nadir footprint, and $Tb(\theta)$ is Tb at any given $\theta$ (Moradi et al., 2015a). Since the SAPHIR data do not suffer from scan asymmetry, we preferably used the satellite data to develop the limb-correction technique. As shown in Figure 2, the limb-darkening is stronger for the channels operating near the center of the water vapor absorption line than channels operating near the wings of the line. Figure 2 also shows the values for the coefficient $c$ in Equation 2 for different SAPHIR water vapor channels.

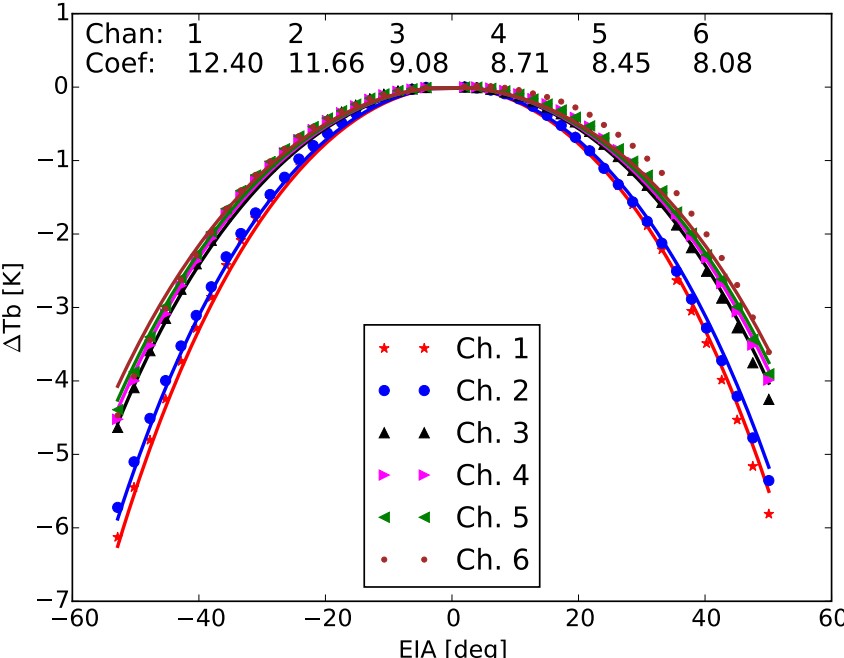

**Figure 2.** Limb-darkening effect as a function of earth incidence angle (EIA) for different SAPHIR channels. The empirical coefficient for Equation 2 are also printed on the plot.

Microwave satellite data are less sensitive to clouds than IR data, however microwave measurements may also be affected by optically thick clouds. Since the empirical coefficients are only valid for clear-sky radiances, the data affected by clouds should be excluded from the analysis. We used the same thresholds proposed by Moradi et al. (2015b) to screen-out the clouds using the differences between Tb's of an upper channel (Tb2, channel 2 operating at $183\pm1.10$ GHz) and a lower channel (Tb5, channel 5 operating at $183\pm6.8$ GHz). The satellites Tb's are cloud free if $Tb2-Tb5 < -15$ K and Tb2 $> 240$ K. More details are provided in Moradi et al. (2015b).

In addition to clouds, some of the satellite radiances may be affected by the surface emissivity which results in high (low) Tb values over land (ocean). Because of the inverse relation between the RH and TB, this corresponds to artificially low (high) RH values over land (ocean) for the measurements that are affected by the surface. We used a threshold for Tb's to exclude the measurements that are affected by the surface. Because the emissivity in microwave frequencies is low over ocean (0.5-0.7) and high over land (about 0.9), the Tb's affected by the surface are normally high over land and low over ocean. We used a subset of Atmospheric Radiation Measurement Program (ARM) radiosonde data and also radiative transfer calculations to determine the thresholds for Tb's that are affected by either land or ocean. We performed two set of radiative transfer calculations using the same radiosonde profiles but different emissivity values for land and ocean. Figure 3 shows the Tb's histograms for different SAPHIR channels, when the difference between simulated Tb's for land and ocean is less than 0.01 K. Therefore, the histograms show the range of Tb's that are not affected by the surface. Based on these histograms we defined the following thresholds for excluding surface affected observations from analysis for channels 1-6, respectively: 230 K to 270 K, 240 K to 280 K, 250 K to 290 K, 255 K to 295 K, 265 K to 295 K, and 270 K to 300 K. The lowest threshold is applied when the observations are affected by the sea surface and the maximum applies when the observations are affected by the land surface. No filter is applied for topography, thus the results over mountainous terrains should be considered with caution since the satellite radiances are averaged over a spatially very inhomogeneous region.

## 3.2 Analyzing Diurnal Variation

Fourier series are traditionally used to model the diurnal cycle of meteorological variables such as temperature and humidity. Fourier series are periodic functions expressed in terms of sine and cosine functions as follows:

$$F(x) = a_0 + \sum_{k=1}^{K} [a_k \cos(kx) + b_k \sin(kx)] \tag{3}$$

where x represents time of the day in radians ($x \in [-\pi, \pi]$) and can be calculated as $x = \frac{t-t_1}{t_1}\pi$, where $t_1$ is equal to 12, $t$ is time of the day in hour, and $a_0$, $a_k$, and $b_k$ are the Fourier coefficients that can be calculated using the following relations:

$$
\begin{aligned}
a_0 &= \frac{1}{\sum w_i} \sum w_i y_i \\
a_k &= \frac{2}{\sum w_i} \sum w_i y_i \cos(kx_i) \\
b_k &= \frac{2}{\sum w_i} \sum w_i y_i \sin(kx_i)
\end{aligned}
\tag{4}
$$

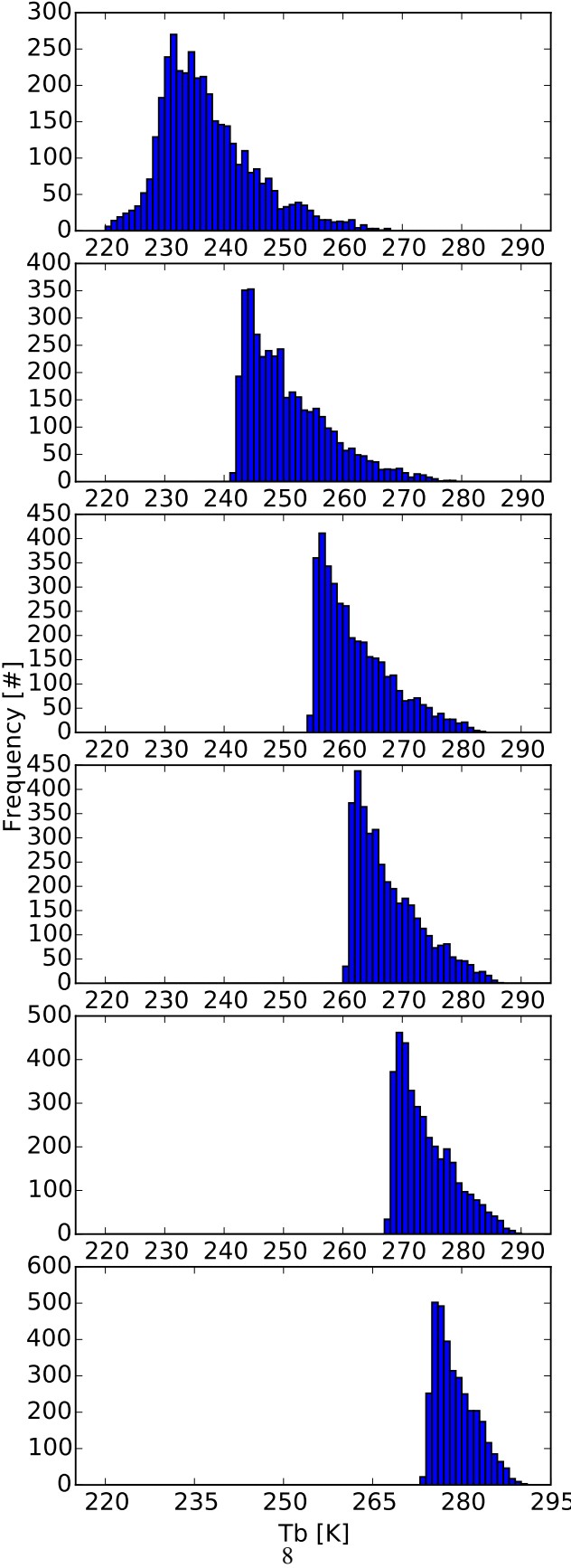

**Figure 3.** Histograms for the distribution of Tb's that are not affected by the surface. The panels from top to bottom are for SAPHIR channel 1 (upper troposphere) to channel 6 (lower troposphere), respectively.

where $y_i$ are the measurements and $w_i$ indicates the weights given to each measurement. The wights are calculated as $\frac{1}{\sigma}$, where $\sigma$ is the standard deviation of all the measurements within each individual bin. We assume that the data are equally spaced and the points define the middle of each interval, so that $n$ data points can be used to divide the space $[-\pi, \pi]$ into $n$ equal intervals so that $\Delta x = \frac{2\pi}{n}$. Some of the previous studies, e.g., Tian (2004) and Kottayil et al. (2013), have used least square techniques to determine the Fourier coefficients. However, as shown in Equation 4, the coefficients can be mathematically and directly calculated form the measurements. It is also required to determine number of terms that Fourier series should be expanded (i.e., $K$ in Equation 3). However, there is no standard method to determine this number. We evaluated the mean absolute difference between the measurements and the values calculated using Fourier series to determine number of terms that Equation 3 needed to be expanded (see Section 4.4 for more information). It was found that one term is sufficient over regions with small diurnal variation, but the series need to be expanded two terms to properly cover the diurnal variation over regions with a larger diurnal amplitude.

## 4 Results

### 4.1 Tb to RH Transformation

We used a subset of the ARM radiosonde data to calculate the empirical coefficients ($a$ and $b$) for Equation 1. Since the saturated vapor pressure can be calculated with respect to either liquid (temperatures above the freezing point of water) or ice phase (temperatures below the freezing point of water), the empirical coefficients can be defined the same way with respect to saturated vapor pressure over either liquid or ice. We use $\mathrm{RH}_I$ to refer to RH with respect to ice and $\mathrm{RH}_L$ for RH over liquid. It is expected that at least in the middle and upper troposphere (channels 1-4), the air temperature is generally below the freezing point thus we need to use the saturated vapor pressure over ice. Additionally, for the lower channels (channels 5 and 6) the saturated vapor pressure expressions for ice and liquid approach each other. Therefore, in most cases we only present the results for the ice phase ($\mathrm{RH}_I$) and the results for the liquid phase ($\mathrm{RH}_L$) are provided in supplementary materials.

Figure 4 shows an example of the relation between satellite Tb's and natural logarithm of layer averaged $\mathrm{RH}_I$ for SAPHIR channel 2. Channel 2 of SAPHIR operates at 183±1 GHz which is similar to a channel on many humidity sounders such as Advanced Technology Microwave Sounder (ATMS), AMSU-B, and MHS. Therefore, the results can be directly compared with the previous studies. For instance, the coefficients shown in Figure 4 are consistent with Buehler and John (2005), Moradi et al. (2010), and Moradi et al. (2015a).

The empirical coefficients for all the channels are presented in Table 1 with respect to both liquid and ice. Coefficient $a$ for Channel 1 over both ice and liquid is smaller than the same coefficient for

**Table 1.** The empirical coefficients for the Tb to RH transformation method for SAPHIR channels.

| Chan. | liquid | | ice | |
| --- | --- | --- | --- | --- |
| | $b$ | $a$ | $b$ | $a$ |
| 1 | -0.059621 | 13.065021 | -0.067173 | 15.231791 |
| 2 | -0.072363 | 16.974748 | -0.080434 | 19.281791 |
| 3 | -0.063765 | 15.758799 | -0.071711 | 18.022020 |
| 4 | -0.061421 | 15.623266 | -0.069159 | 17.818025 |
| 5 | -0.060818 | 16.033315 | -0.069916 | 18.581239 |
| 6 | -0.061955 | 16.826082 | -0.072675 | 19.818404 |

other channels, but all other coefficients are very close for all the channels. Both $a$ and $b$ coefficients are greater over ice than over liquid.

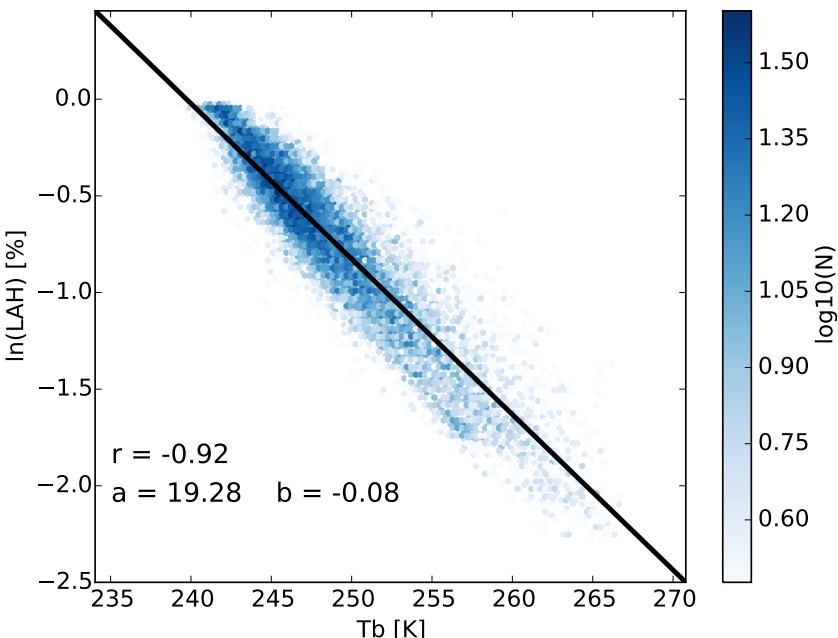

**Figure 4.** Relation between satellite Tb and natural logarithm of layer-averaged $RH_I$. The colorbar shows the logarithm of the number of observations.

## 4.2   Spatial Distribution of RH

SAPHIR observations are available for the tropical region expanding from about 35 S to 35 N. However, we limited the study to the region between 25 S to 25 N, because the frequency of the revisits
is limited outside this region. We first binned the data based on local observation time into a grid

of $1.0 \times 1.0$ degree. The data-points within each grid-box were gridded into 24 bins based on local observation time. The local time was calculated using Coordinated Universal Time (UTC) and longitude which are both provided in SAPHIR data. Finally, we averaged the data within each grid-box of $1.0\,^\circ \times 1.0\,^\circ \times 1.0\,\text{hr}$. Figure 5 shows average number of observations per hour after the data are fil-

tered for clouds and surface effect. See section 4.3 and Table 2 for details on the boxes shown on the maps. Since the satellite inclination is about $20°$, maximum number of observations occurs around $10\,°\text{N}$ to $20\,°\text{N}$ and $10\,°\text{S}$ to $20\,°\text{S}$. As shown, on average, 100 to 300 observations are retained for each bin per hour. Over very high elevations the weighting functions for all the channels peak very close to the surface, therefore, the minimum number of observations occurs over mountains such as

the Andes in South America.

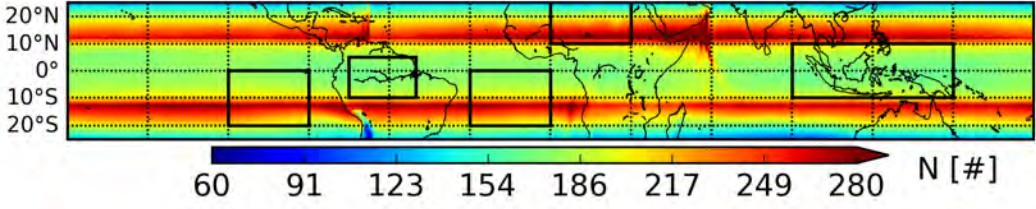

**Figure 5.** Mean number of overpasses per hour for SAPHIR channel 6. Colorbar shows number of observations per grid point. Number of overpasses are generally higher for other channels than for channel 6.

Figure 6 shows the mean daily $\text{RH}_L$ for different SAPHIR channels. As expected, RH significantly changes from upper troposphere ($14\,\%$ to $36\,\%$) to lower troposphere ($41\,\%$ to $85\,\%$ for Channel 6). Note that in order to avoid the outliers especially over the Andes, in most cases, the upper limits printed on the colorbars show the ninety-ninth percentile. The dry regions are observed over sub-

sidence regions, e.g., South Pacific Ocean, South Atlantic Ocean, as well as Arabian Sea, and are consistent with previous studies (e.g., Moradi et al., 2010; Eriksson et al., 2010; Chung et al., 2007). Additionally, several moist regions are observed over convective regions, e.g., South America, Central Africa, and South Asia also known as Maritime Continent which is located within Tropical Warm Pool. The RH generally decreases with distance from the convective regions which is con-

sistent with previous studies. For instance, Udelhofen and Hartmann (1995) reported that the upper tropospheric humidity rapidly decreases with distance from the convective clouds. The pattern does not change from upper to lower troposphere, but generally the decrease in moisture with distance from the convective region, especially over the Maritime Continent, is faster in lower troposphere than in upper troposphere. It is also evident that moist regions are connected so that water vapor can

be transported across the Equatorial region.

Figure 7 shows the layer-averaged tropospheric $\text{RH}_I$. Because the saturated vapor pressure is lower over ice than over liquid, the $\text{RH}_I$ values are greater than the $\text{RH}_L$ values. So that $\text{RH}_I$ ranges between $18\,\%$ to $52\,\%$ in upper troposphere (compared to $14\,\%$ to $36\,\%$ for $\text{RH}_L$), and $38\,\%$ to $90\,\%$

in lower troposphere (compared to $41\%$ to $85\%$ for $\mathrm{RH}_L$). Since we excluded the data that are affected by the surface, no difference between land and ocean is observed along the coastlines. The Andes show a large impact on the lower tropospheric RH but a small impact on upper tropospheric RH. The results for regions such as the Andes should be interpreted with cautious because of uncertainty in the measurements over mountainous regions. The fact that the results do not show a significant land/sea contrast shows that the filter for the surface effect properly removes the surface affected observations. The values reported in Figure 6 are about $10\%$ to $15\%$ higher than the values reported in Chung et al. (2007). As stated before, this is because the moist regions are removed from the IR observations due to strict cloud screening. This is also consistent with John et al. (2011) who reported about $10\%$ dry-bias in clear-sky IR observations in the upper troposphere.

Polar orbiting satellites orbit the earth twice a day, thus the daily average of relative humidity estimated from the measurements of these satellites may be biased depending on the crossing time. In order to evaluate the impact of crossing time on the estimated daily averages, we used the observations from only two overpasses being 12 hours apart, similar to ascending and descending orbits of polar-orbiting satellites. We then computed the mean difference of the daily averages calculated using only two overpasses and the daily averages calculated using all the hourly data. Figure 8 shows the results for the measurements from 01:30 (13:30) local time and Figure 9 shows the results for 09:30 (21:30) local time. In fact we collected all the measurement 30 minutes before and after the aforementioned local time. The mid-nigh/afternoon orbit (01:30/13:30 local time) matches with several satellites including NOAA Joint Polar Satellite System and NASA A-Train, and the early morning/late evening orbit (09:30/21:30 local time) matches with the orbit for the MetOp satellites. As shown, especially for the lower tropospheric channels, the error is generally larger for 01:30/13:30 local time, because as shown later, in tropical region the peak of RH generally happens in early morning over many regions. In both cases, the error is generally less than $2\%$ $\mathrm{RH}_I$ in upper troposphere and slightly increases in middle troposphere. However, in lower troposphere the difference between the two cases is considerable. The error due to eliminating diurnal variation is more than $4\%$ $\mathrm{RH}_I$ especially over land for the measurements from 01:30/13:30 local time and overall less than $2\%$ $\mathrm{RH}_I$ for the measurements from 09:30/21:30 local time. These results show that measurements from polar-orbiting satellites can be used to derive the mean tropospheric humidity in the tropical region with a good accuracy. However, polar-orbiting satellites may not provide a good picture of peak time and amplitude, because these parameters show a large spatial inhomogeneity and obviously depend on the satellite overpass time (see Section 4.3 for more details). The same figures are included in the supplementary materials for the error in $\mathrm{RH}_L$.

## 4.3 Amplitude and Peak Time

Figure 10 shows the diurnal amplitude of layer-averaged $RH_I$ as the difference between maximums and minimums of RH derived from the Fourier series fit over the course of the day. We employ the

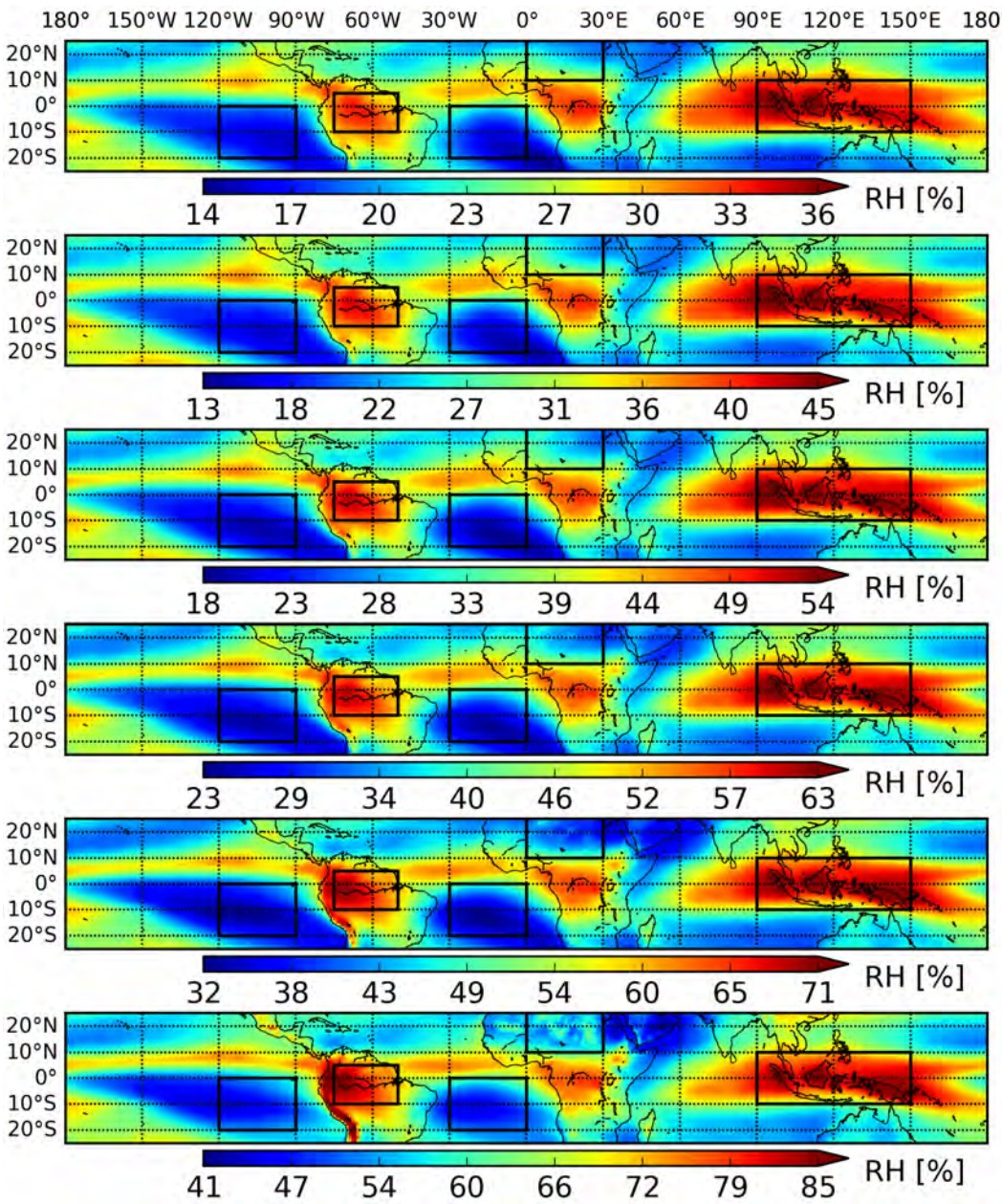

**Figure 6.** Spatial distribution of layer-averaged $RH_L$ derived using SAPHIR data for the period January 2012 to September 2015. Depicts from top to bottom are for SAPHIR channels 1-6, respectively.

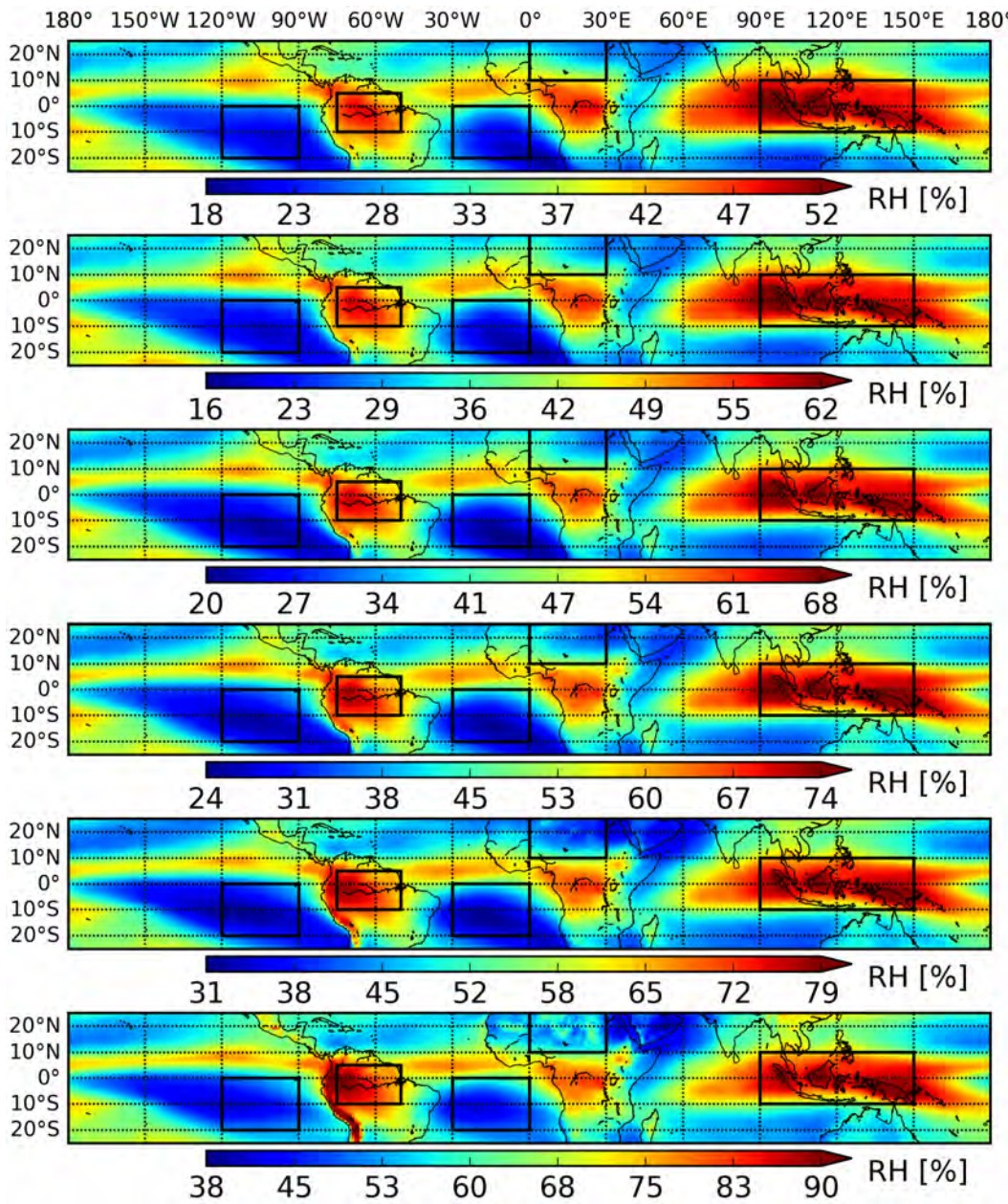

**Figure 7.** Spatial distribution of layer-averaged RH$_I$ derived using SAPHIR data for the period January 2012 to September 2015. Depicts from top to bottom are for SAPHIR channels 1-6, respectively.

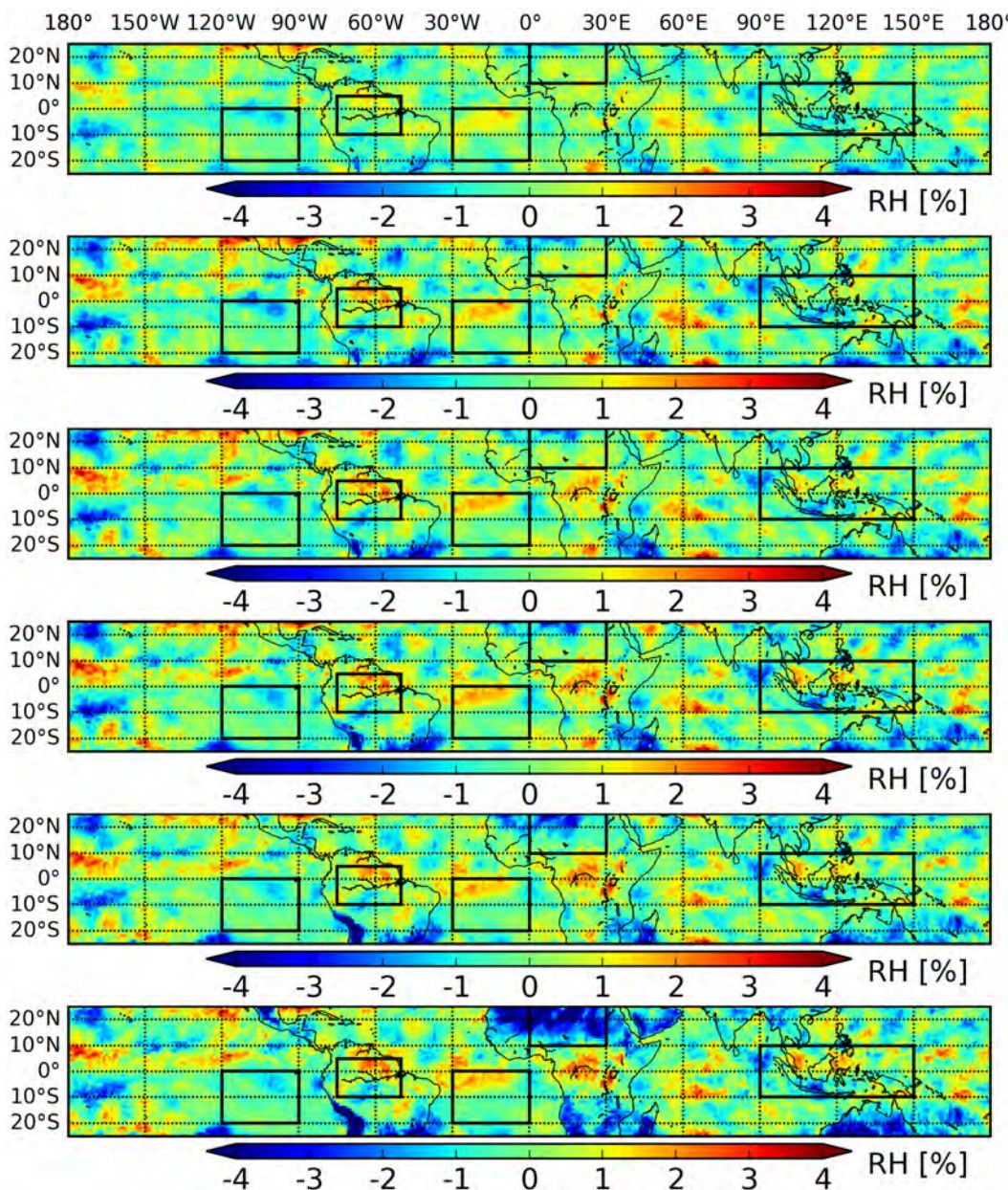

**Figure 8.** Mean difference of daily average of RH$_I$ calculated using only data from 01:30/13:30 local time and the daily average calculated using all hourly data. Depicts from top to bottom are for SAPHIR channels 1-6, respectively.

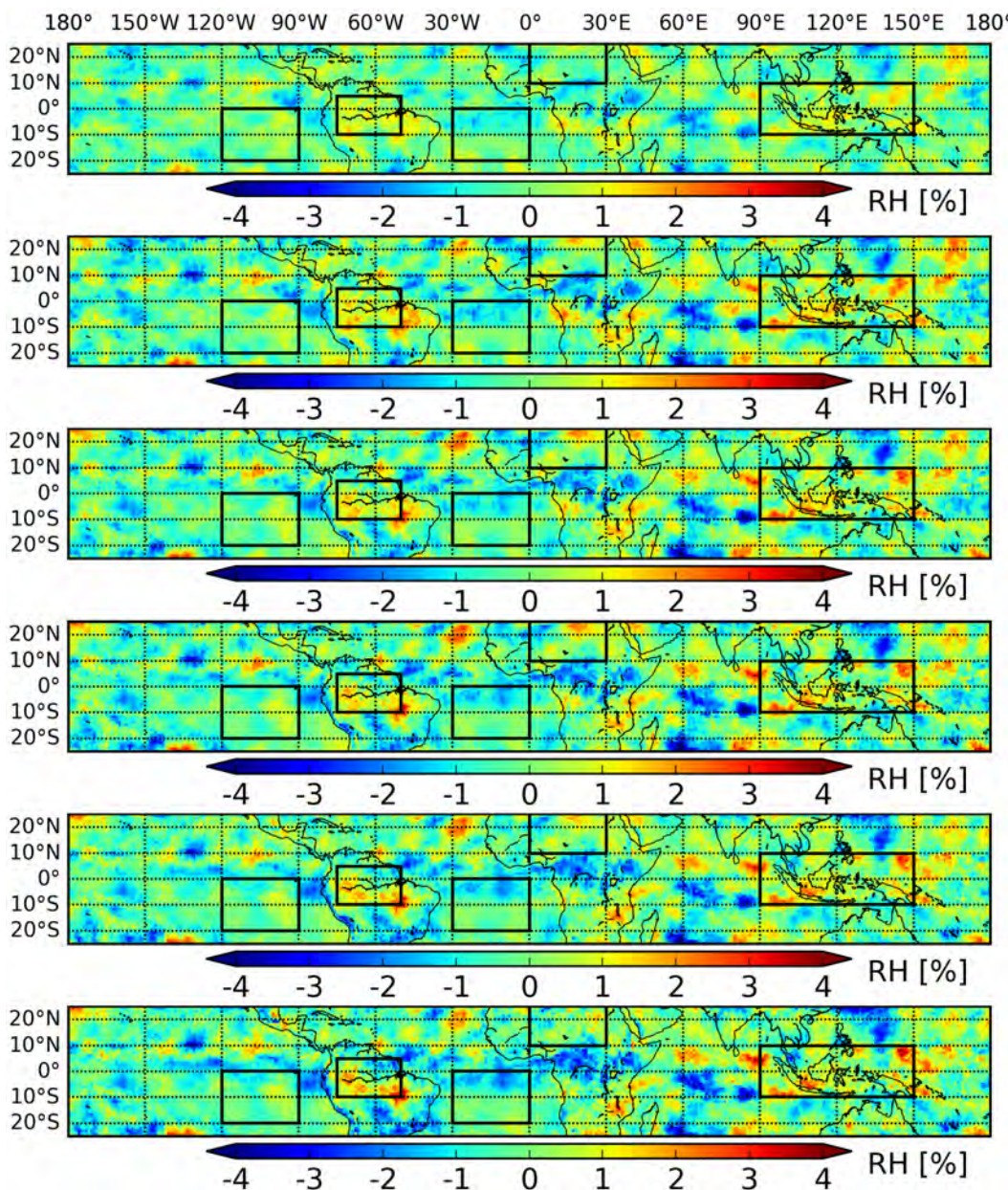

**Figure 9.** Mean difference of daily average of $RH_I$ calculated using only data from 09:30/21:30 local time and the daily average calculated using all hourly data. Depicts from top to bottom are for SAPHIR channels 1-6, respectively.

diurnal amplitude and peak time derived from the Fourier series, because they are more stable and less noisy than the amplitude and peak time derived from the measurements, though both Fourier series and measurements yield very similar results. The diurnal amplitude derived from the measurements is included in the supplementary materials. The pattern and magnitude of the amplitude significantly change from upper to lower troposphere. In upper and middle troposphere, i.e., Chan-

nels 1-4, the diurnal amplitude is less than 15 % with the maximum occurring over the Andes, South Africa, Madagascar, and Australia as well as some scattered places over South America and Arabian Desert. The diurnal amplitude in upper troposphere is consistent with Eriksson et al. (2010) who reported up to 8 % change over tropical land regions in upper troposphere. However, in lower troposphere, the amplitude can be up to 29 % over land. In lower troposphere, the diurnal amplitude

is generally less than 10 % over ocean, but over deserts and mountains the amplitude is greater than 15 % with maximum occurring over deserts of South and North Africa, Australia, Middle East, the Andes in South America, and the Sierra Madres mountains in Mexico. The diurnal amplitude for $RH_L$ (see the supplementary materials) is a few percent smaller than that for $RH_I$ but the pattern is very similar. The amplitude slightly changes from channel 1 to channel 5, but it is much larger

for channel 6 than for channel 5 (29 % versus 19 % over land). However, the pattern is very similar for channels 5 and 6. Generally, a stronger diurnal amplitude is found over land than over ocean for the lower channels. This can be explained by the fact that the diurnal variation of RH in the lower layers of the troposphere is enforced by the change in surface and boundary layer temperature which are larger over land than over ocean. Over ocean, the diurnal amplitude tends to be larger over con-

vective regions than over subsidence regions which is consistent with previous studies (e.g., Chung et al., 2007).

Since the location of ITCZ changes with time, the diurnal amplitude and peak time are expected to be seasonal dependent. The ITCZ location is furthest aways from the equator during Summer Solstice (towards north) and Winter Solstice (towards south), therefore, in addition to examining

the annual averages of the diurnal amplitude and peak time, we separately evaluated the diurnal amplitude and peak time for the months of December and January (Winter Solstice, Figure 11) as well as June and July (Summer Solstice, Figure 12). Comparing the diurnal amplitude of RH derived from the annual averages (Figure 10) with Figures 12 and 11 shows that the change in diurnal amplitude during Summer Solstice is larger than the change during the Winter Solstice. This

is because due to the presence of more continents in the northern hemisphere, ITCZ moves further away from the equator during Summer Solstice (Waliser and Gautier, 1993). The main difference between the diurnal amplitudes of Winter and Summer Solstices can be summarized as follows. The amplitudes in all layers are about 2 % higher in Winter Solstice than in the Summer Solstice. Shift in ITCZ forces the convective region over Central Africa to move to the southern latitudes during

Winter Solstice. Therefore, North Africa is dominated by a dry subsidence region during Winter Solstice which causes a larger diurnal amplitude especially in the lower troposphere likely due to a

larger diurnal cycle in air temperature. In contrast, in summer Solstice, ITCZ moves to the northern latitudes so that most African continent is dominated by a convective region, except South Africa, which causes a smaller diurnal amplitude during Summer Solstice. In addition, the diurnal amplitude over North America is considerably lower during Summer Solstice than the Winter Solstice.

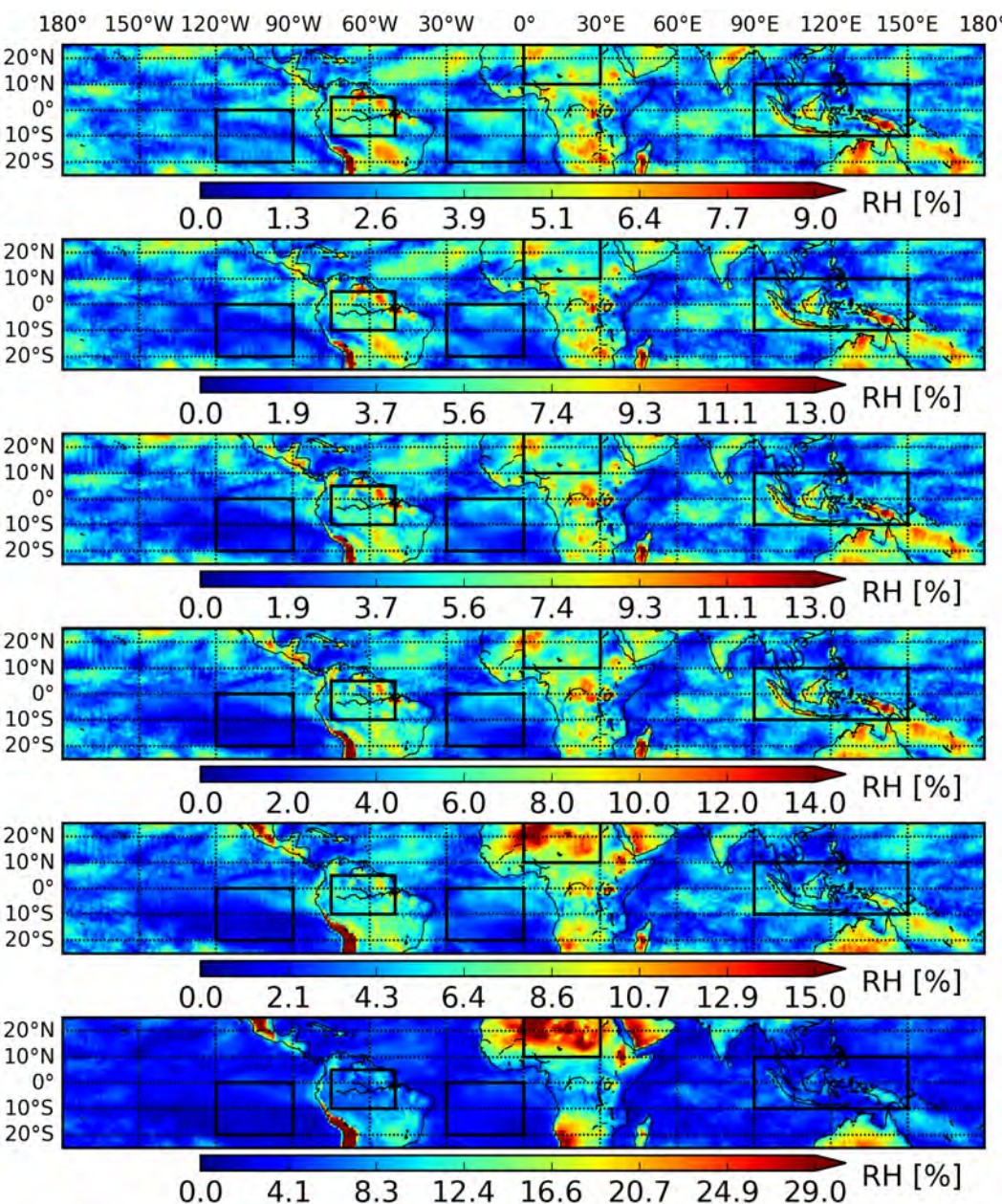

**Figure 10.** Spatial distribution of diurnal amplitude of $RH_I$. Depicts from top to bottom are for SAPHIR channels 1-6, respectively.

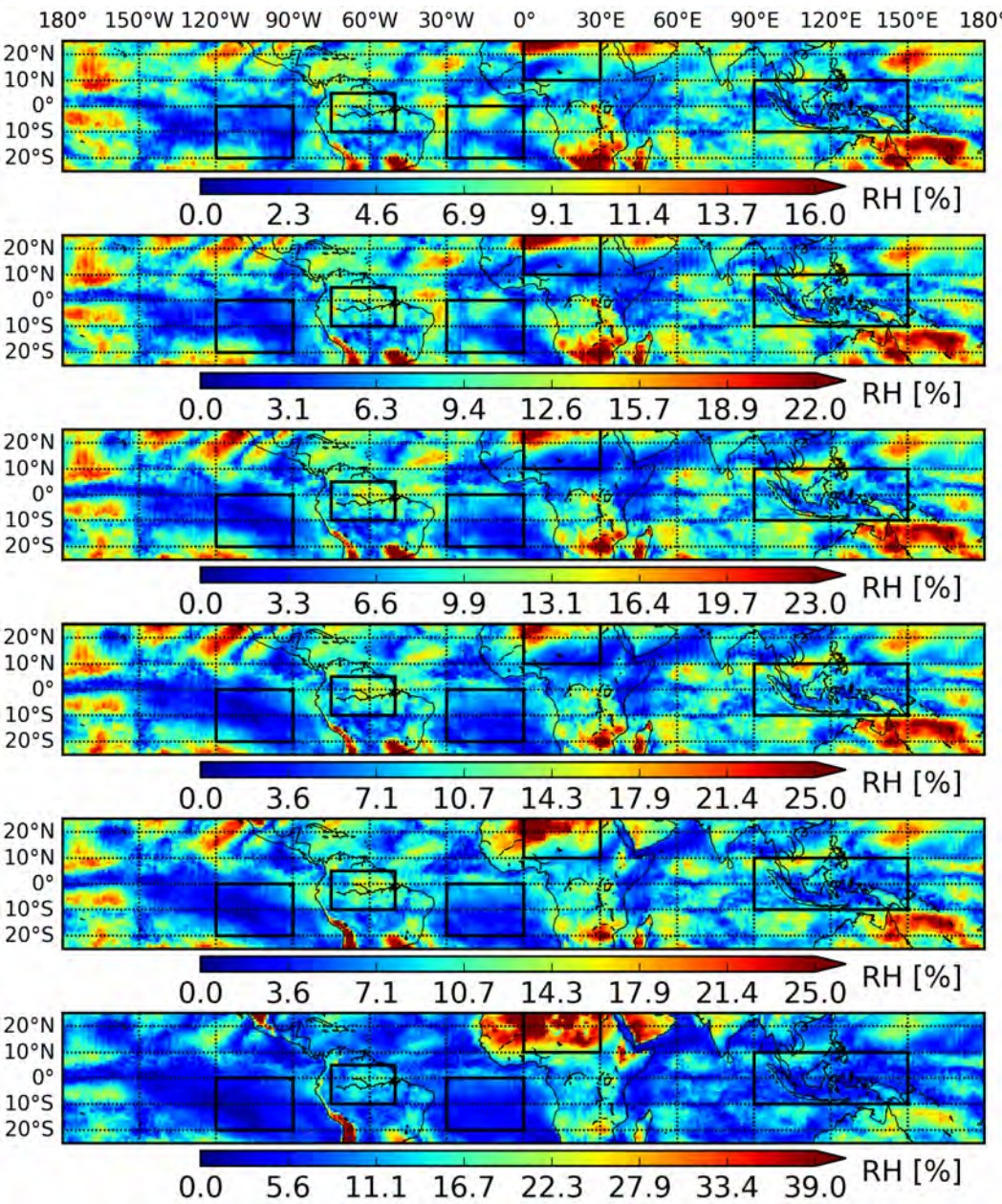

**Figure 11.** Spatial distribution of diurnal amplitude of $RH_I$ for the months of December and January. Depicts from top to bottom are for SAPHIR channels 1-6, respectively.

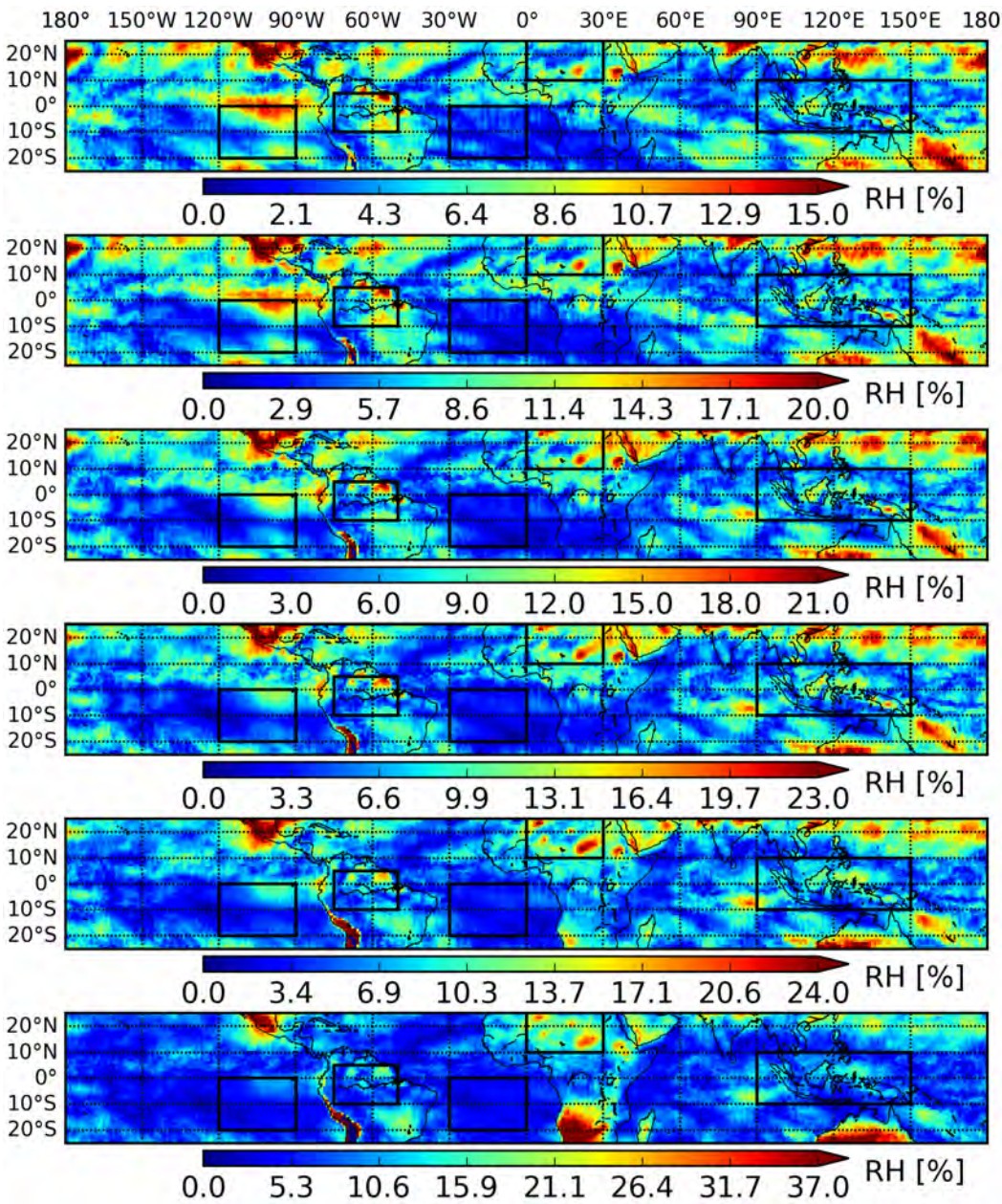

**Figure 12.** Spatial distribution of diurnal amplitude of $RH_I$ for the months of June and July. Depicts from top to bottom are for SAPHIR channels 1-6, respectively.

Figure 13 shows the $RH_I$ diurnal peak time for different channels derived from the Fourier series fit. The peak time derived from the measurements is shown in the supplementary materials. Generally, over most regions, the peak time is delayed from upper to lower troposphere. For instance, over Africa the diurnal peak time changes from midnight in the upper troposphere to early morning in the lower troposphere. Overall, the peak-time occures around mid-night over most continental regions which is consistent with previous studies (e.g., Chung et al., 2007). Over oceans, the peak-time normally happens in the morning over subsidence regions but mid-night over the convective regions. It should be noted that there is a large uncertainty in estimating the peak time when the diurnal amplitude is very small. Generally, the early morning peak time that has been reported before (e.g., Soden, 2000; Tian, 2004; Chung et al., 2007; Eriksson et al., 2010; Kottayil et al., 2013), only occurs in some regions and it is not common for the entire tropical region. The early morning peak time is due to a peak in deep convective activities (Haffke and Magnusdottir, 2015; Albright et al., 1985). In upper troposphere, the nightitme peak over land is generally consistent with Chung et al. (2007), however Chung et al. (2007) reported a late-afternoon/evening peak time over the convective regions which is not consistent with the current study. This can be due to the fact that the peaktime derived from IR data is affected by the cloud screening method. Channel 3 in Kottayil et al. (2013) (middle panel in Figure 5) can be directly compared with channel 2 in this study. Over South America (Amazonian region) both studies show night-time peak time. However, over some other regions, e.g., Indian Ocean, the results are not consistent. We show a peak time before 00:00 LT over most part of Indian Ocean but Kottayil et al. (2013) reported a night-time peak (between 00:00 and 02:00 LT). We have filtered cloud contaminated data, therefore the results are not dominated by cloud diurnal regime in the tropical region. Since Kottayil et al. (2013) did not exclude the surface affected data, the results for the lower channels cannot be compared.

Haffke and Magnusdottir (2015) reported a large inhomogeneity in the diurnal variation of precipitation which is consistent with the inhomogeneity we have found for relative humidity. Using Tropical Rainfall Measuring Mission (TRMM) Precipitation Radar measurements, Biasutti et al. (2011) reported a large difference between diurnal variation of precipitation over coastal lands (early afternoon peak) and near shores (early morning) influenced by the land/sea breeze. These fine structures cannot be detected in MW observations due to coarse spatial resolution, but it highlights the fact that the results over inhomogeneous regions such as coastal regions may not be very representative. Although, the spatial resolution of the SAPHIR observations do not provide a sharp transition from ocean to land, a distinct peak time is observed along some of the shorelines. For instance, on the west coast of South America, the peak time over lands near shore occurs a few hours earlier than the peak time for both inland and free waters.

Figures 14 and 15 shows the peak-time for Winter and Summer Solstices respectively. The most distinguishable difference between the peak-times during Winter and Summer Solstices are as follows. Over Eastern Pacific, in the middle and lower troposphere, the peak mostly happens in the early

morning during Summer Solstice but late evening and midnight during Winter Solstice. A nighttime peak is observed over Africa during Summer Solstice but the peak-time changes to early morning during Winter Solstice. Over Indian Ocean, there is an early morning peak-time during Summer Solstice that changes to a late-evening and midnight peak during Winter Solstice.

**Figure 13.** Diurnal peak time in local time for $RH_I$ based on Fourier series fit. Depicts from top to bottom are for SAPHIR channels 1-6, respectively.

Based on the mean and diurnal amplitude of RH, five regions are selected that will be used to further investigate some of the results. The coordinates (minimum and maximum of latitudes and longitudes) of these rectangular regions are shown in Table 2. These regions from west to east are located over South Pacific Ocean, Amazon, South Atlantic Ocean, North Africa, and the Maritime Continent. These regions are selected in a way to present a high diversity in diurnal amplitude as well as mean tropospheric RH. The boundaries for these regions are shown on all the maps.

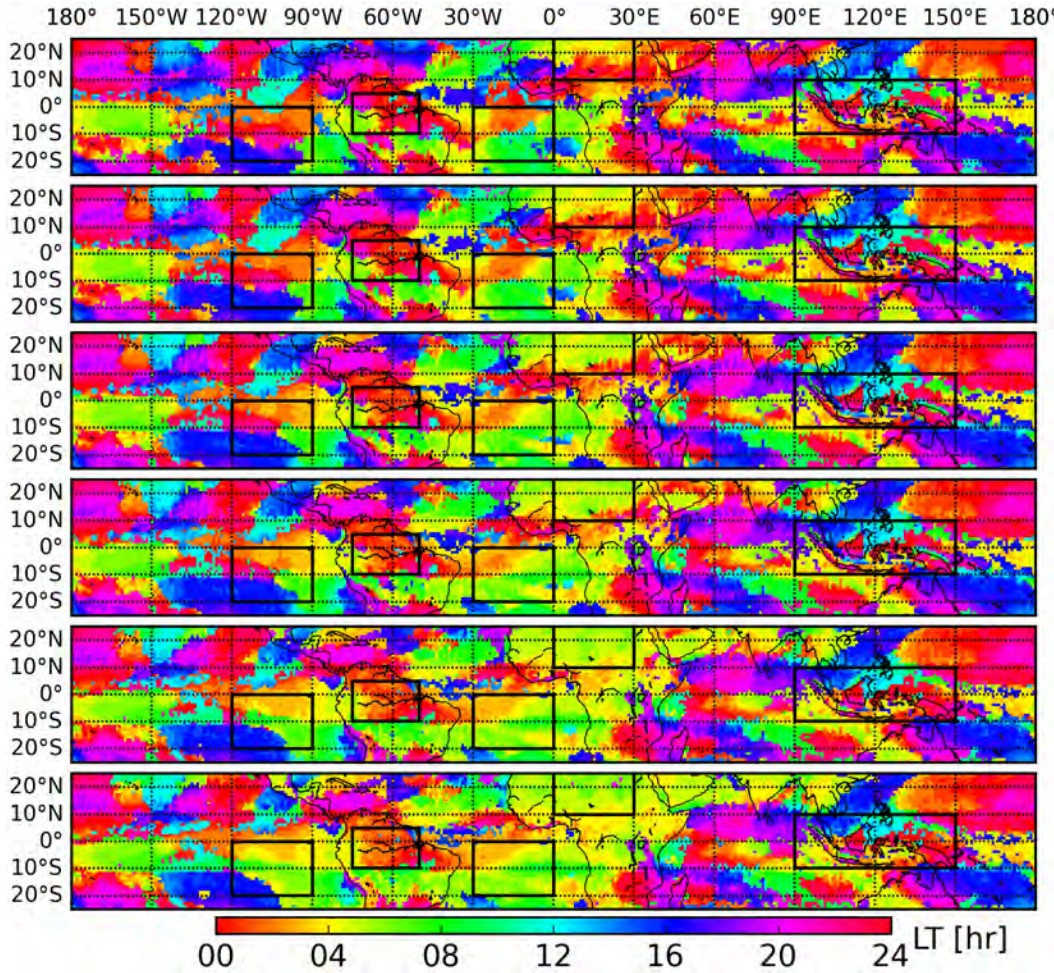

**Figure 14.** Diurnal peak time in local time for RH$_I$ based on Fourier series fit for the months of December and January. Depicts from top to bottom are for SAPHIR channels 1-6, respectively.

**Table 2.** The regions selected based on mean and amplitude of tropospheric RH to investigate the diurnal variation of RH. The regions are labeled from west to east as South Pacific Ocean (SP), Amazon (AM), South Atlantic Ocean (SA), North Africa (NA), the Maritime Continent (MC), and entire Tropical Region (TR). These regions are indicated on all the maps.

| Label | Lat1 | Lat2 | Lon1 | Lon2 |
|-------|------|------|------|------|
| SP | -20 | 0 | -120 | -90 |
| AM | -10 | 5 | -75 | -50 |
| SA | -20 | 0 | -30 | 0 |
| NA | 10 | 25 | 0 | 30 |
| MC | -10 | 10 | 90 | 150 |
| TR | -25 | 25 | -180 | 180 |

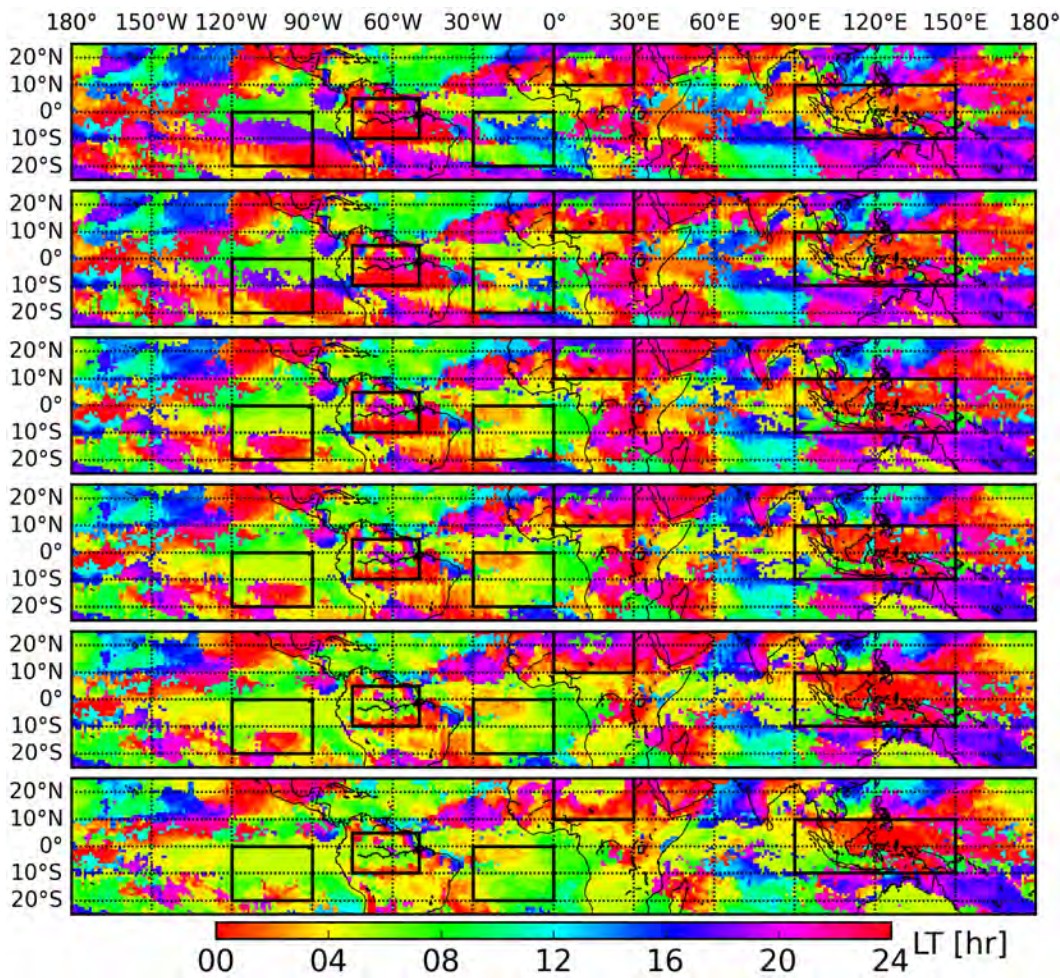

**Figure 15.** Diurnal peak time in local time for $RH_I$ based on Fourier series fit for the months of June and July. Depicts from top to bottom are for SAPHIR channels 1-6, respectively.

## 4.4 Diurnal Cycle of RH

The diurnal cycle of tropospheric RH is modeled using Fourier series and the series themselves can be demonstrated using the Fourier coefficients. As mentioned before, Fourier series can be theoretically expanded by infinite terms but in practice only a few terms are required for diurnal variation of meteorological variables. In this study, we expanded the Fourier series in only two terms, i.e., k=1,2. As mentioned earlier, this number was determined by analyzing the difference between the measurements and the values estimated using Fourier series. For instance, Figure 16 shows the mean absolute difference between the measurements and the Fourier series fit. Over most regions the difference is less than one percent when the series are expanded in two terms.

Since the coefficient $a_0$ is equal to the mean of the measurements, its distribution is already shown in Figures 6 and 7. The coefficient $a_k$ and $b_k$ present information about the phase and amplitude of

the signal and are included in the supplementary materials. If we rewrite the Fourier coefficients as a
complex number $z_n = a_n + ib_n$ then the amplitude and phase can be expressed using $|z|$ and $arg(z)$:

$$|z| = \sqrt{zz*} = \sqrt{a^2 + b^2} \tag{5}$$

$$arg(z) = x = tan^{-1}(\frac{b}{a}) \tag{6}$$

where, $arg(z)$ is the phase difference ranging from $-\pi$ to $\pi$ and can be converted back to the time
of the day using $t = (12x/\pi) + 12$.

In the following, the selected regions will be used to further investigate the diurnal variation of
tropospheric RH using Fourier series. Figure 17 shows the diurnal cycle of the layer-averaged RH$_I$
(absolute values) in selected regions for different SAPHIR channels. Figure 18 shows the RH values
scaled between $-100$ and $100$ to better distinguish the diurnal cycle for regions with a small diurnal
amplitude. As shown, in most regions the diurnal amplitude is very small (less than a few percent),
but the diurnal amplitude in middle and lower troposphere over North Africa is greater than $10\%$
(consistent with Figure 10). In upper troposphere (Channels 1 and 2), the diurnal variation is less than
$5\%$ over all selected regions. As shown in Figure 18, some regions such as SP and SA experience
two peaks. Over SP, one of the peaks happens in early morning and other one in the evening. The
evening peak is stronger in upper troposphere, but in lower troposphere the morning peak becomes
stronger than the evening peak. A similar pattern exist over SA, however the second peak occurs
around early afternoon and becomes weaker from upper to lower troposphere. However in both cases
the actual amplitude of the diurnal cycle is a very small percentage. The double peak is consistent
with a double peak in precipitation reported in Dai (2001). Note that peak in precipitation normally
coincide with a minimum in RH. As reported in Chung et al. (2007) the RH trend, at least in upper
troposphere, follows the trend in convective clouds except the minimum of RH that occurs when the
precipitation starts to increase. As shown in Figure 17, the peak time slightly shifts from upper to
lower troposphere. The amplitude over North Africa also changes significantly from a few percent
in upper troposphere to more than $20\%$ in lower troposphere which is due to change in the diurnal
variation of air temperature. Figure 17 also includes an example for a grix-box with a large difference
between the measurements and the fit for the Fourier series. The grid-box is located near Lago Salar
de Arizaro ($24.5\,°$S, $67.5\,°$W), a small salt flat of the Andes in Argentina. The surface area of the flat
is only $1600\,km^2$, therefore the grid-box covers a mix of the salt flat and the surrounding terrains. As
shown, the Fourier series are perfectly fitted to the data and the difference between the measurements
and the Fourier series fit is rather due to noise in the data.

## 4.5 Distribution Functions

Figure 19 shows the distribution of layer averaged RH with respect to both liquid and ice over
selected regions. In addition, the cumulative distribution functions are also provided in the supple-

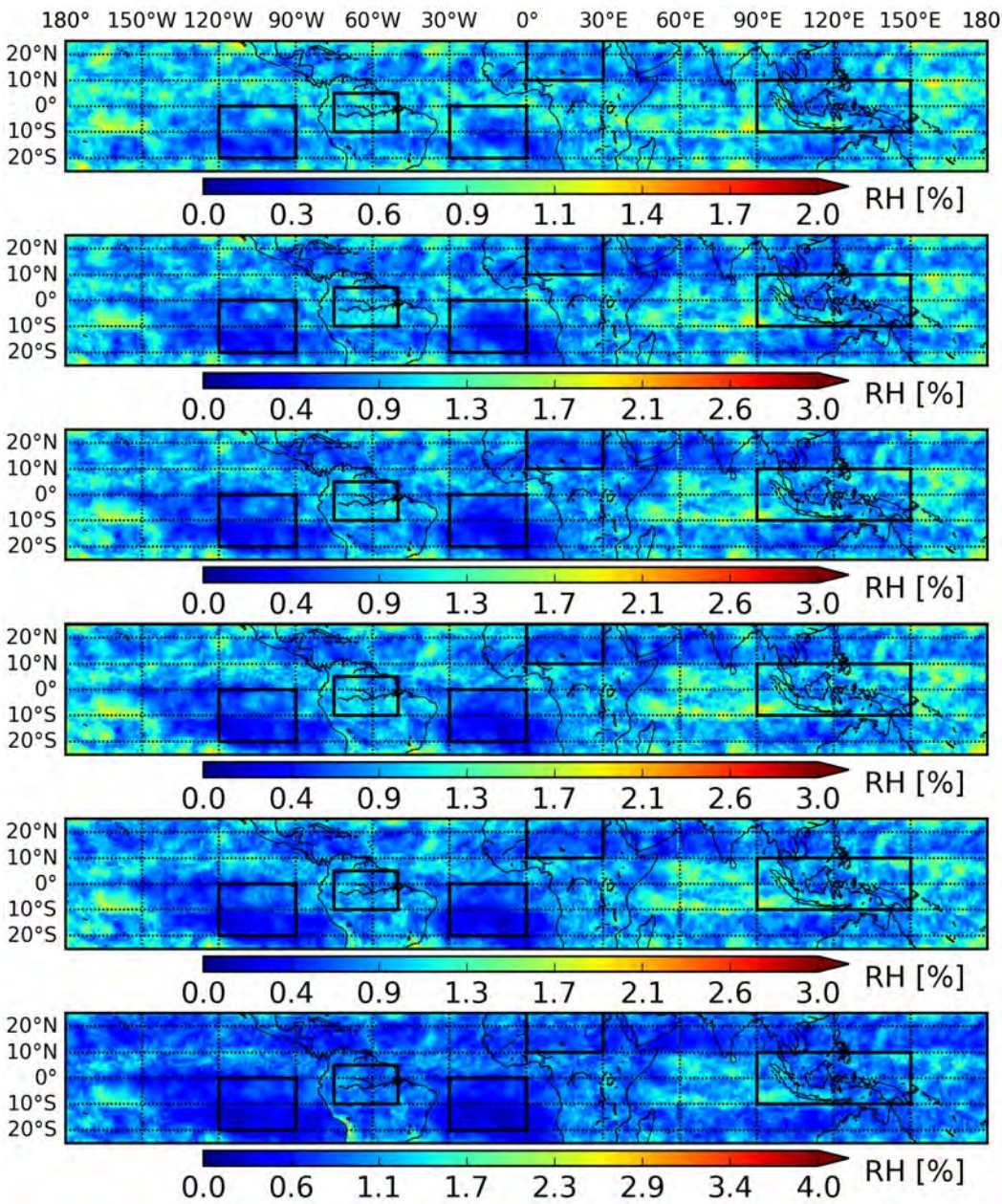

**Figure 16.** Mean absolute difference (with respect to ice) between measurements and the fit for Fourier series. Depicts from top to bottom are for SAPHIR channels 1-6, respectively.

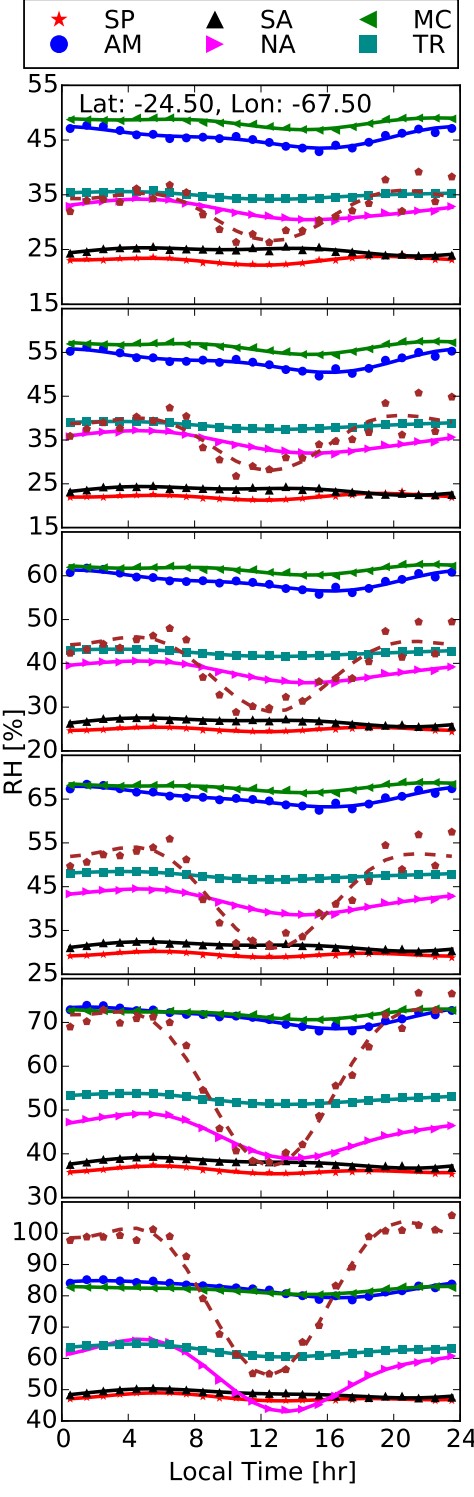

**Figure 17.** Diurnal cycle of layer-averaged RH$_I$ as well as Fourier series fit for the selected regions. Depicts from top to bottom are for SAPHIR channels 1-6, respectively. The legend shows the name of the regions which are defined in Table 2. The dashed lines show a grid box with a large difference between the measurements (pentagons) and the fit for the Fourier series (the dashed-line). The longitude and latitude of this grid-box are printed on the top plot.

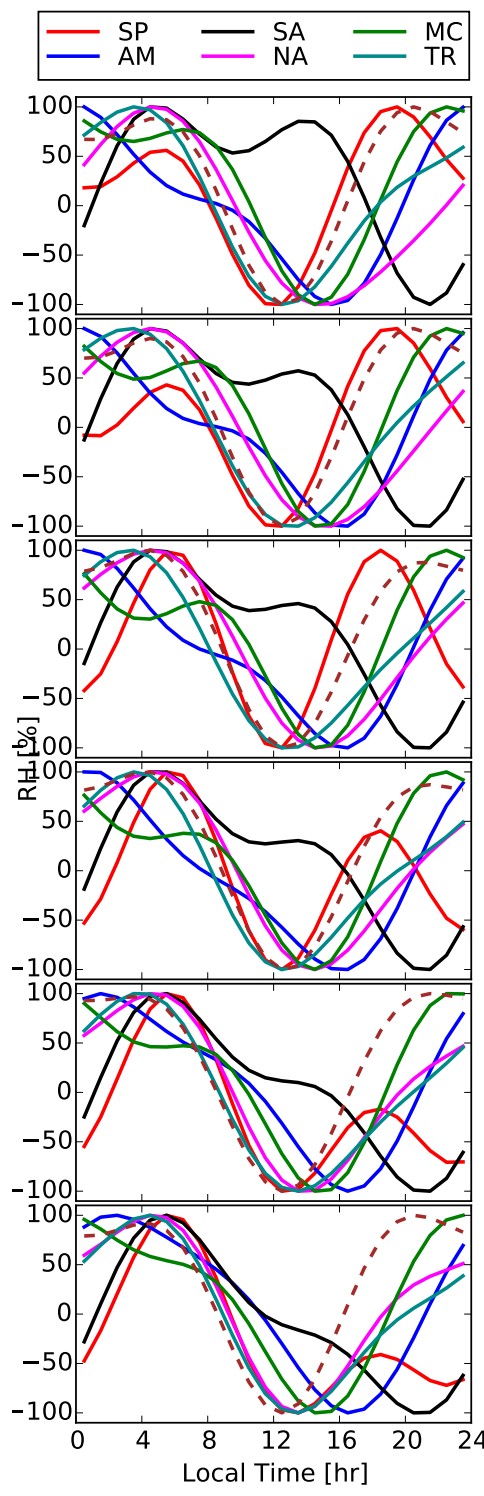

**Figure 18.** Same as Figure 17 but the values are scaled between -100 and 100.

mentary materials. As shown the distributions of RH with respect to ice and water are similar in the lower troposphere (channels 5 and 6). This is because the relative humidity values calculated using Equation 1 are nearly the same over ice and liquid for the expected range of brightness temperatures in lower troposphere. This is an indication that the transition from ice to liquid is performed smoothly. When necessary, we use RH over ice for channels 1-4, and over water for channels 5-6 to discuss the results. Overall, the distribution functions are very similar for the Amazon and the Maritime Continent regions. Some small differences exist between the two regions especially in the upper troposphere, where the Maritime Continent tend to be more moist than Amazon. The distribution functions are also similar for South Pacific and South Atlantic, but South Atlantic tends to be slightly more moist than South Pacific in all layers. Therefore, we only explain distribution functions for South Atlantic, the Maritime Continent, and North Africa in more detail. The first and third quartiles of the distribution functions are used to explain the range of RH in each layer. The range of RH between first and second quartile include 50% of the data-points. For channel 1, the first (third) quartiles are 15(25), 40 (55), 20 (40), over South Atlantic, the Maritime Continent, and North Africa. None of the regions show a normal distribution function. The distribution is left-skewed (the left tail is longer) for Amazon and the Maritime Continent, and right-skewed over the rest of the selected regions. The shape of the distribution remains almost similar for other channels. The quartiles are very similar between channels 1 and 2, but increase about 5% per layer for channels 3-5. So that the quartiles for each lower channel are about 5% higher than the same quantities for the channel peaking above it. From channels 5 to channel 6, the increase in quartiles is about 10%. Note that in many cases the minimum and maximum are very close to first and third quartile meaning that 25% of the datapoints lie within a small range of RH. A small percentage of the data show supersaturation over liquid (up to $110\%$) for Channel 6. This can be explained by the methodological error as well as error in the satellite observations. We estimate that the error introduced by difference sources (see Section 4.6) can be up to $15\%$. Our findings for SAPHIR Channels 1 and 2 are generally consistent with Eriksson et al. (2010), especially for the results presented for AURA-MLS instrument. For instance both studies show that the $RH_I$ can reach up to $80\%$ over Africa. However, Eriksson et al. (2010) reported a maximum $RH_I$ of $80\%$ to $100\%$ over the Maritime Continent but our results show a maximum of $60\%$ to $80\%$ for SAPHIR Channel 1 and $80\%$ to $100\%$ for SAPHIR Channel 2. Overall, it is expected that the cloud filter removes some supersaturated regions in our study, because those regions are normally associated with the cloud formation.

### 4.6 Error Estimates

In this section, the error sources are discussed, though it is not possible to quantitatively estimate most of the errors. One obvious source of error is bias and noise in the satellite data. According to Moradi et al. (2015b) the bias in SAPHIR data should be less than $0.5\,\text{K}$ which is roughly equal to $5\%$ in RH space. Although, the bias affects the RH values, it does not affect the results for diurnal

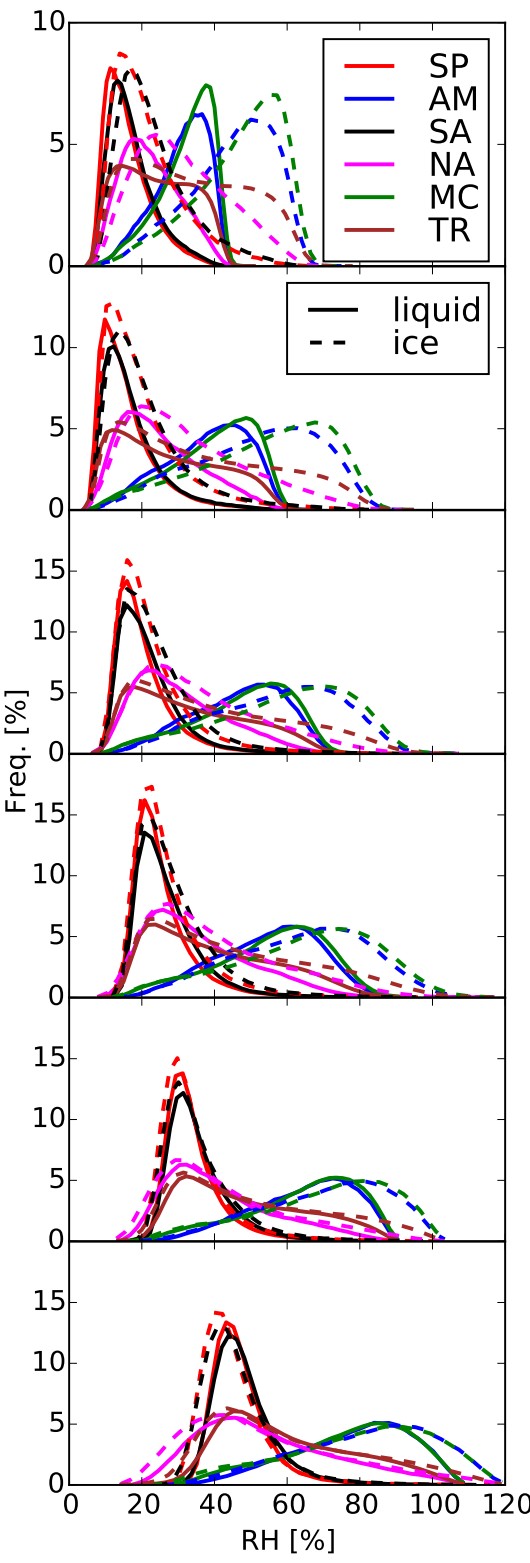

**Figure 19.** Distribution functions for both $RH_I$ and $RH_L$. Depicts from top to bottom are for SAPHIR channels 1-6, respectively. The legend shows the name of the regions which are defined in Table 2.

amplitude and peak time. Since a large volume of the data is averaged, the noise in the satellite data
should cancel out if the relation between RH and Tb was linear. However, due to the non-linear
relation between the two, the random noise may not completely cancel out. Another source of error
is from the Tb to RH transformation method which according to Moradi et al. (2015a) is estimated
to be less than 10 %. Other sources of error include the surface and cloud effects. Although we
have applied appropriate filters, it is still possible that some clouds are not filtered out and at least
for the lower channels there are still cases that are affected by the surface. The limb-correction
technique is based on satellite data averaged over the entire tropics. Therefore, it may introduce
at least some noise in the correction for the individual measurements. However, it is expected that
the limb-correction does not introduce a systematic bias. The random error introduced by the limb-
correction technique may not completely cancel out due to a non-linear relation between RH and
Tb. However, the overall impact of the random error on the results is expected to be negligible.
Finally, diurnal variation of RH is highly influenced by the diurnal variation of air temperature, thus
sources such as inversion in temperature lapse rate can contribute to the error because satellite data
are averaged over a wide layer.

## 5 Conclusions and Summary

Water vapor significantly influences the Earth's climate because of its greenhouse effect. Water vapor
is also important to the global water and energy budget. Free tropospheric water vapor, especially in
the tropical region, significantly contributes to the water vapor feedback through radiative processes
(Trenberth et al., 2009; Dessler et al., 2008). Diurnal cycle of humidity influences diurnal vari-
ations in other geophysical variables such as precipitation, and convective activities (Chung et al.,
2007). Despite the importance of water vapor and its diurnal variation, current climate and numerical
weather prediction models do not adequately simulate the diurnal variation of tropospheric humidity
(Chung et al., 2013). As the simulations of climate models are improved, more accurate observations
are required to verify the validity of simulations. Most studies, so far have used IR measurements
to investigate changes in tropical tropospheric RH, but because of high sensitivity of IR channels
to clouds, these studies are significantly biased toward analyzing dry conditions as cloud screening
methods remove moist regions from the analysis. Some previous studies have used multi-instrument
microwave observations to study diurnal cycle of RH, but due to inter-satellite differences there is
a large uncertainty in such studies. SAPHIR is a microwave instrument onboard Megha-Tropiques
that provides frequent observations in the tropical region with frequent daily revisits.

In this study, we first transformed the satellite Tb's into layer-averaged RH, then binned the
SAPHIR data using the location and local observation time into a grid of $1.0\degree \times 1.0\degree \times 1.0\,\text{hr}$. Finally,
we fitted the Fourier series to the data within each grid-box. The daily-averaged RH values showed
that the moist regions are associated with the convective regions and the dry regions are associated

with the subsidence regions. The results reported in this study for mean tropospheric humidity are

10 % to 15 % higher than those reported using IR observations (e.g., Chung et al., 2007) which is because the moist regions are removed from the IR observations due to cloud screening. In upper and middle troposphere the amplitude is generally less than 15 %, but in lower troposphere the amplitude can reach up to 30 %. The diurnal amplitude is generally smaller over ocean than over land and over ocean it tends to be larger over convective regions than over subsidence regions which is consistent

with previous studies (e.g., Chung et al., 2007). An early morning peak-time was observed for most tropical band, but there are several regions where the peak-time occurs over night or in the afternoon. The early morning peak is due to a peak in convective activities and is generally consistent with previous studies, however new results show that the early morning peak-time is not common in the entire tropical region. The diurnal amplitude and peak time slightly change during the year due

to shift in ITCZ. A double peak (one peak in the morning and one in the afternoon) is observed over some regions of the tropics which is consistent with double peak reported for precipitation (e.g., Dai, 2001). We sampled SAPHIR observations similar to polar-orbiting satellites (twice a day) to investigate the impact of sampling on the estimated tropospheric humidity. The results showed that although polar orbiting satellite may estimate the mean tropospheric humidity with a good accuracy,

they may not be able to estimate diurnal amplitude and peak-time with enough accuracy especially in the lower troposphere.

The results were analyzed separately for RH with respect to saturated vapor pressure over both ice and liquid. The results for both phases are either included in the paper or in the supplementary materials. The analysis shows that microwave measurements from low-inclination satellites are a

valuable source for investigating the diurnal and spatial variation of tropospheric RH. The application of the current data is limited to RH as these measurements are most sensitive to the RH than absolute humidity parameters such as specific humidity. The microwave temperature imaging instrument on the same satellite failed shortly after the satellite was launched, otherwise it would have been possible to perform the same analysis for the tropical tropospheric temperature.

**Appendix A:  Fourier Series**

Fourier coefficients for any known function can be calculated using the following relations:

$$a_0 = \frac{1}{2\pi} \int\limits_{-\pi}^{\pi} f(x)dx$$

$$a_k = \frac{1}{\pi} \int\limits_{-\pi}^{\pi} f(x)\cos(kx)dx \qquad\qquad (A1)$$

$$b_k = \frac{1}{\pi} \int\limits_{-\pi}^{\pi} f(x)\sin(kx)dx$$

The relations presented in Equation A1 are useful when the Fourier series are used to approximate a known function but in our specific case, $f(x)$ is unknown and need to be approximated. These relations can be discretized based on a Riemann sum and the measurements of RH as follows:

$$a_0 = \frac{1}{2\pi} \sum_{i=1}^{n} y_i \Delta x = \frac{1}{n} \sum_{i=1}^{n} y_i = \bar{y}$$

$$a_k = \frac{1}{\pi} \sum_{i=1}^{n} y_i \cos(kx_i)\Delta x = \frac{2}{n} \sum_{i=1}^{n} y_i \cos(kx_i) \qquad\qquad (A2)$$

$$b_k = \frac{1}{\pi} \sum_{i=1}^{n} y_i \sin(kx_i)\Delta x = \frac{2}{n} \sum_{i=1}^{n} y_i \sin(kx_i)$$

     Equation A1 can be rewritten as shown in Equation 4 when different weights are given to the measurements.

*Acknowledgements.* This study was supported by NOAA grant# NA09NES4400006 (Cooperative Institute for Climate and Satellites - CICS) at the University of Maryland, Earth System Science Interdisciplinary Center
(ESSIC). Part of the research was carried out at the Jet Propulsion Laboratory, California Institute of Technology, under a contract with the National Aeronautics and Space Administration. SAPHIR data are processed and provided by Centre National d'Etudes Spatiales (CNES), France. The views, opinions, and findings contained in this report are those of the authors and should not be construed as an official National Oceanic and Atmospheric Administration or U.S. Government position, policy, or decision.

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
