# Peer review of "Diurnal variation of tropospheric relative humidity in tropical region"

_Atmospheric Chemistry and Physics, 2015_

## Referee Comment (RC1) · Anonymous Referee #1 · 11 Feb 2016

**General comments**

The authors used brightness temperature data from 6 channels in the 183 GHz water vapour absorption line, measured with the SAPHIR instrument on Megha-Tropiques, to infer the diurnal cycle of layer averaged relative humidity in the tropics, up to $25°$ from the equator. The 6 channels correspond to 6 thick layers in different altitudes from the bottom of the free troposphere to the upper troposphere. The use of microwave (MW) data from this instrument is a progress relative to earlier attempts using infrared (IR) data or data from polar orbiting satellites, because MW is much less sensitive to clouds and the low inclination angle of Megha-Tropiques allows to sample the tropics much more often than just twice daily.
[Figure]

It turns out that the diurnal cycle is weak over most regions, but with exceptions where the amplitude can be very large (as the pentagons curve in Figure 13 exemplifies). The daily peak occurs mostly in the early morning, but there are places where it occurs in the afternoon as well. Interestingly, there seems to be no correlation between the daily amplitude and the general level of relative humidity.

To my opinion, the paper is a useful and interesting contribution to ACP. The following lists minor points. It should be no problem to address them.

**Minor comments**

Lines 20-34: The motivation for the study is centred around the many roles of water vapour (WV) in the climate system. However, the study is about the diurnal cycle of free tropospheric WV. How are the subtleties of WV's diurnal cycle related to climate issues, is there a connection at all? To my opinion it is important to study the diurnal cycle since it exists. I believe, however, that the diurnal cycle is not important for the climate issues and thus the motivation may not be appropriate. Perhaps the authors can provide arguments for such a link.

Lines 74/75: Unclear sentence. Which T change is meant? Do you mean that the diurnal RH variation is correlated with the diurnal T-variation? A reference for this statemant would be fine.

Lines 85-87: On first reading I had the impression, motivated by the $x\pm y$ error bar style, that the channels get broader and broader from ch. 1 to ch. 6. But this is probably wrong. I believe now that $183 \pm 11$ GHz is not a 22 GHz wide channel, but a channel which detects radiation at 172 and 194 GHz. Probably this is meant with "double pass band". Could you please clarify this?

Line 89: check whether 6 km is correct; appears too low.

Lines 90/91: The variation of the Jacobian's peaks with moisture content of the atmosphere is a problem in the IR, too. Is the problem particularly strong in the MW region?

Eq. 1: Please specify the meaning of upper and lower indices (probably channel and swath position). Note that it is mathematically incorrect to have lower indices $i$ on the rhs of the equation, but not on the lhs. Did you forget a sum sign?

Line 124: check use of "upper" and "lower". Do these words refer to the channel number or to the peak altitude of the respective Jacobian?

Lines 127-143: I do not really understand what you describe here. First, how the channels with high peak altitude can be influenced by the surface. Regarding figure 1, the Jacobians of chs. 1 and 2 should be close to zero at the ground. Or is this contamination from high mountains? Second, why there is a cut-off at both low and high temperatures? How does the surface feign a Tb below 230 K for instance? Additionally, it would help, if the figure would not only show data, that are NOT affected by the surface, but also those data, that are affected. Does the warning for mountainous terrain not imply that data over land are generally bad, since there is mountainous terrain everywhere on land in the tropics? Can we only trust the data over ocean?

Eq. 4: Where do the weights come from and how are they determined?

Line 173: Whether RH is expressed as $RH_I$ or $RH_L$ is independent of whether an ice phase exists or not. The justification for preferring $RH_I$ is unnecessary.

Lines 223-242: The authors estimate errors due to insufficient temporal sampling of polar orbiting satellites here. It seems, that these errors are generally small, both at single locations (Figures 8 and 9) and on average. One should note that it is hardly possible to measure RH better than to about 10% in RH. Compared to this typical error margin, 2-4% difference is little. Also when I compare these differences with the diurnal amplitude (Figure 10) or with the difference of measurements vs. Fourier fit (Figure 12),

it seems that they can almost be neglected. Please comment on this.

Line 331: I suggest to write "distribution of layer averaged RH". It is important here to distinguish between the local RH (which is usually understood under the term RH) and the layer averaged, non-local, RH. Since these data are already layer averaged, extreme values are largely smoothed away. Distributions based on local data would be much broader then what is shown in Figure 14.

Lines 333-335: There is no saturation pressure with respect to ice above zero (Celsius). How did you calculate the RHI for these lower channels? And does the sentence "we use RH over ice ..." not contradict the line types in Figure 14 (dashed and solid)?

Figure 14: I understand ice supersaturation in the figure, but it looks as if there were water supersaturation as well. Please check and if there is water supersaturation try to explain.

Line 366: what is the transformation method?

Line 375: which change in air temperature is meant?
* * *

---

## Author Comment (AC1) · 18 Feb 2016

**1   Reviewer #1**

We sincerely appreciate the reviewer for providing very helpful and constructive comments on the paper. We basically agree with all the comments and will consider them when revising the manuscript. In order to make benefit of the peer-review interactive discussion, we are providing our answers to the reviewer's comments as well as the changes that we will make in the revised version of the manuscript.

*Lines 20-34: The motivation for the study is centred around the many roles of water vapour (WV) in the climate system. However, the study is about the diurnal cycle of free tropospheric WV. How are the subtleties of WV's diurnal cycle related to climate issu23es, is there a connection at all? To my opinion it is important to study the diurnal cycle since it exists. I believe, however, that the diurnal cycle is not important for the climate issues and thus the motivation may not be appropriate. Perhaps the authors can provide arguments for such a link.*

As pointed out by the reviewer, we realized that we have not explicitly discussed the importance of diurnal cycle of humidity in climate. We added the following sentences to clarify this:

> Diurnal cycles in temperature and moisture drive diurnal variations in temperature, precipitation, and convective activities (Chung et al., 2007), therefore, are expected to interact significantly with, for example, changes in global mean humidity or temperature. However, current climate and numerical weather prediction models do not adequately simulate the diurnal variation of tropospheric humidity (Chung et al., 2013), a failing that is very likely to lead to inaccuracies in their simulations. As models are improved, accurate observations of diurnal cycles of humidity will be crucial in verifying the validity of simulations.

*Lines 74/75: Unclear sentence. Which T change is meant? Do you mean that the diurnal RH variation is correlated with the diurnal T-variation? A reference for this statement would be fine.*

We amended the text as follows to better reflect the meaning:

As stated before, relative humidity is the ratio of water vapor pressure to the saturated vapor pressure. The water vapor pressure depends mainly on the water vapor content of the atmosphere, but the saturated vapor pressure depends on the air temperature. Therefore, the diurnal variation of RH does not necessary indicate change in the water vapor content of the atmosphere, because it is affected by both diurnal variation of water vapor and air temperature. For instance, change in the amount of lower tropospheric water vapor over deserts is very small during day, but RH can significantly change because of change in air temperature. Therefore, it is more desired to analyze the diurnal variation of absolute humidity parameters.

Lines 85-87: On first reading I had the impression, motivated by the x±y error bar style, that the channels get broader and broader from ch. 1 to ch. 6. But this is probably wrong. I believe now that 183±11 GHz is not a 22 GHz wide channel, but a channel which detects radiation at 172 and 194 GHz. Probably this is meant with "double pass band". Could you please clarify this?

Yes, as the reviewer stated, the x±y is conventionally used to specify the double passbands not the error bars

Line 89: check whether 6 km is correct; appears too low.

We thanks the reviewer for noting this. The correct number is about 10 km. We have amended the text accordingly.

Lines 90/91: The variation of the Jacobian's peaks with moisture content of the atmosphere is a problem in the IR, too. Is the problem particularly strong in the MW region?

Yes that is correct. The peak altitude of both IR and MW Jacobians change with the humidity. We have amended the text to reflect this.

Eq. 1: Please specify the meaning of upper and lower indices (probably channel and swath position). Note that it is mathematically incorrect to have lower indices i on the rhs of the equation, but not on the lhs. Did you forget a sum sign?

We removed the index $i$ as it was not necessary and have explained that $ch$ stands for the channels. The $a$ and $b$ coefficients are calculated separately for each channel so no summation is required (we may have misunderstood the comment on adding a sum sign!).

Line 124: check use of "upper" and "lower". Do these words refer to the channel number or to the peak altitude of the respective

Many thanks to the reviewer for noting the error. We have reworded the sentence as follows:

> We used the same thresholds proposed by Moradi et al., 2015, to screen-out the clouds using the differences between Tb's of an upper channel (Tb2, channel 2 operating at 183±1.10 GHz) and a lower channel (Tb5, channel 5 operating at 183±6.8 GHz).

Lines 127-143: I do not really understand what you describe here. First, how the channels with high peak altitude can be influenced by the surface. Regarding figure 1, the Jacobians of chs. 1 and 2 should be close to zero at the ground. Or is this contamination from high mountains? Second, why there is a cut-off at both low and high temperatures? How does the surface feign a Tb below 230 K for instance? Additionally, it would help, if the figure would not only show data, that are NOT affected by the surface, but also those data, that are affected. Does the warning for mountainous terrain not imply that data over land are generally bad, since there is mountainous terrain everywhere on land in the tropics? Can we only trust the data over ocean?

The reviewer is correct and the measurements for high peaking channels are normally not affected by the surface. However, it is very likely that over high mountains the measurements for those channels are also affected by the surface. We have already explained in the text that the results over mountains should be used with caution. The Jacobians are plotted for all the profiles regardless of whether there is any surface effect or not. Overall, the surface emissivity for water vapor channels is about 1 over land and 0.5-0.7 over ocean. Thus, the brightness temperatures affected by the surface are high over land and low over ocean. Therefore, we need a filter that removes both high and low brightness temperatures. In the case of mountains, the surface emissivity is close to one but since the land surface temperature is low, the affected brightness temperatures may not be rejected by the surface filter.

Eq. 4: Where do the weights come from and how are they determined?

We amended the text as follows to explain how the weights are calculated:

> The wights are calculated as $\frac{1}{\sigma}$, where $\sigma$ is the standard deviation of all the measurements within each individual bin.

Line 173: Whether RH is expressed as RHI or RHL is indepen-
dent of whether an ice phase exists or not. The justification for
preferring RHI is unnecessary.
We rewrote the whole paragraph as follows to explicitly state that we
mean saturated vapor pressure above and below freezing points:

> We used a subset of the ARM radiosonde data to calculate the empir-
> ical coefficients ($a$ and $b$) for Equation 1. Since the saturated vapor
> pressure can be calculated with respect to either liquid (temperatures
> above the freezing point of water) or ice phase (temperatures below
> the freezing point of water), the empirical coefficients can be defined
> the same way with respect to saturated vapor pressure over either
> liquid or ice. We use $\mathrm{RH}_I$ to refer to RH with respect to ice and $\mathrm{RH}_L$
> for RH over liquid. It is expected that at least in the middle and up-
> per troposphere, the air temperature is generally below the freezing
> point. Additionally, for the lower channels the saturated vapor pres-
> sure expressions for ice and liquid approach each other. Therefore,
> in most cases we only present the results for the ice phase ($\mathrm{RH}_I$) and
> the results for the liquid phase ($\mathrm{RH}_L$) are provided in supplementary
> materials.

Lines 223-242: The authors estimate errors due to insufficient
temporal sampling of polar orbiting satellites here. It seems,
that these errors are generally small, both at single locations
(Figures 8 and 9) and on average. One should note that it is hardly
possible to measure RH better than to about 10% in RH. Compared to
this typical error margin, 2-4% difference is little. Also when I
compare these differences with the diurnal amplitude (Figure 10)
or with the difference of measurements vs. Fourier fit (Figure
12), it seems that they can almost be neglected. Please comment on
this.
We agree with the reviewer that over most regions these differences can
be counted as negligible. We added the following sentence to clarify this:

> Overall, over most regions, these differences are small and can be
> considered negligible compared to the methodological errors.

Line 331: I suggest to write "distribution of layer averaged
RH". It is important here to distinguish between the local RH
(which is usually understood under the term RH) and the layer av-
eraged, non-local, RH. Since these data are already layer averaged,
extreme values are largely smoothed away. Distributions based on
local data would be much broader then what is shown in Figure 14.

Done!

There are in fact two independent equations for saturated vapor pressure over ice and liquid that require air temperature as input. The equation for the saturation over liquid (ice) is just more accurate for temperatures above (below) the freezing point. We already amended the text to state that we mean saturation vapor pressure for the temperatures above/below freezing point (because of the super cold water) rather than the actual liquid/ice phase. We also amended the text as follow to avoid any confusion:

> When necessary, we use RH over ice for channels 1-4, and over water for channels 5-6 to discuss the results

Thanks to the reviewer for reminding this. Yes, a small percentage of data show supersaturation even over liquid (up to 110%) that can be explained by the methodological uncertainty as well as some possible errors in the data (all together probably around 15% error). Please see the section on "error estimates" for more information on the possible sources of errors. We also added the following sentence in the revised version to clarify this:

> A small percentage of the data show supersaturation over liquid (up to 110 %) for Channel 6. This can be explained by the methodological error as well as error in the satellite observations. We estimate that the error introduced by difference sources (see Section 4.6) can be up to 15%.

We amended the text as
> the Tb to RH transformation method

We amended the text as
> the diurnal variation of air temperature

---

## Referee Comment (RC2) · Anonymous Referee #2 · 25 Feb 2016

Comments on

"Diurnal variation of tropospheric relative humidity in tropical region" by Isaac Moradi et al.

Anonymous Referee #2

General comments: The paper addresses an interesting topic, it attempts to analyse the diurnal cycle of humidity in tropical regions in a more coherent manner with a multi-channel microwave instrument in a drifting orbit. I agree with the authors that there is still scope for better analyses of the diurnal cycle of tropospheric humidity, and the mesurements used in this paper provide a good basis for such an analysis. As the authors state in their abstract: results showed a large inhomogeneity in diurnal variation of tropospheric relative humidity in tropical region. This is not new. However

[Discussion paper]

[Figure]

the authors have not built their analysis on earlier work, as they should do. My second major concern is that the analysis is too descriptive with little attempts to explain the observations in terms of meteorology/physics. Therefore I find the way the analysis is done not acceptable and cannot recommend the paper for final publication. In fact it is rather a reject with an inivitation for a new paper.

Let me develop the reasons for my assessment: - The authors state: Despite the importance of water vapor especially in the tropical region, the diurnal variations of water vapor have not been investigated in the past due to the lack of observations. - This not true. - Let me give you a few examples of papers that analyse the diurnal cycle of upper tropospheric moisture, the first one goes back more than 20 years. And their were even earlier papers in proceedings such as the TOVS conference.

1. Influence of tropical cloud systems on the relative humidity in the upper troposphere, Petra M. Udelhofen,
Dennis L. Hartmann, JGR 1995 !!!, DOI: 10.1029/94JD02826

2. Diurnal variation of upper tropospheric humidity and its relations to convective activities over tropical Africa, E. S. Chung, B.J. Sohn, J. Schmetz and M. Koenig, Atmos. Chem. Phys. Discuss., 7, 351–381, 2007

3. Diurnal variation of outgoing longwave radiation associated with high cloud and UTH changes from Meteosat-5 measurements, Eui-Seok Chung , Byung-Ju Sohn, Johannes Schmetz, Meteorology and Atmospheric Physics, October 2009, Volume 105, Issue 3, pp 109-119

4. Chung, E.-S., B. J. Soden, B. J. Sohn, and J. Schmetz (2013a), An assessment of the diurnal variation of upper tropospheric humidity in reanalysis data sets, J. Geophys. Res. Atmos., 118, 3425-3430, doi:10.1002/jgrd.50345.

- Interestingly the authors make use of the reference Chung et al., 2007, yet without comparing their own work rigorously with the results of Chung et al. - The paper as it stands, presents results without referring to relevant earlier work on the very topic.

- The fact the earlier work mostly used IR data is no 'show stopper' for a rigorous comparison. In fact the diurnal cycle of relative humidity measured with microwave instruments in cloudy regions should be fully in phase with the diurnal cycle of clouds, which in turn can be well observed with IR instruments. So in cloudy regions the microwave observations do not provide much new information on the phase of the diurnal cycle of relative humidity. - The diurnal variability of moisture in clear areas can also be observed with IR instruments, as the authors say. And it can well be compared with results in the current paper. This will be interesting because the diurnal cycle of clear sky moisture is indeed quite variable, as the authors observe.

I summary I cannot recommend publication of the paper in its present form. The main reason is the neglect of earlier work. I am surprised about this neglect given how easy it is today to find relevant papers. And, I am sorry, I also have to recall that this should not happen in a serious scientific analysis.

The data are a good source for an analysis and I recommend that the paper gets completely rewrittten in the light of relevant results from other papers. Please note that it is not sufficient to just work the references I provide above into the paper. There is more relevant work in scientific literature. This means my recommendation is a 'reject in its present form' and a request for a major rewriting.

Another strong suggestion is to interpret the results for the diurnal cycles in different regions in terms of meteorology/physical processes.

I am willing to review a substantially revised paper again. Then I will also provide more detailed comments.
* * *

---

## Referee Comment (RC3) · Anonymous Referee #3 · 3 Mar 2016

This study uses microwave measurements from SAPHIR onboard the Megha-Tropiques satellite to investigate the diurnal variation of water vapor in the tropics. In this study, the limb effect-corrected observed radiances were transformed into layer-averaged relative humidity and then partitioned into 24 bins of local observation time. The authors determined the phase and amplitude of diurnal variation by fitting the Fourier series to the binned data, and showed a large inhomogeneity in the diurnal variation of tropospheric relative humidity in the tropics. Although there are some issues that the authors need to clarify, the results presented in this study appear to help improve our understanding of the diurnal variation of water vapor.

Main points:

1. Motivation of the study: Although the authors argues that the diurnal variation of

water vapor has not been investigated due to the lack of observations, the argument is not true given the previous studies addressed in the manuscript. Also, the diurnal variation presented in the manuscript is very weak over most regions of the tropics, raising a question on the need of investigation on the diurnal variation of water vapor. Moreover, it is unclear in what ways the diurnal variation of water vapor is important.

2. The authors argue that their results are superior to previous studies based on IR measurements or multi-instrument microwave measurements. However, it doesn't seem that the differences from the previous studies are discussed comprehensively in the manuscript. Therefore, it is not certain whether this study advances our understanding of the diurnal variation of water vapor in the tropics.

3. The manuscript documents the peak time and amplitude of the diurnal variation in the tropics, but does not provide reasonable physical mechanisms responsible for the spatial and altitude discrepancies in the diurnal variation. Also, it is unclear why some parts of the tropics have an early morning maximum of the tropospheric relative humidity.

4. Given the seasonal migration of ITCZ, the amplitude and peak time of the diurnal variation for a given season might be different from that for the annual mean. However, this aspect is not investigated in the manuscript.

Specific points:

1. L1-2: The argument that the diurnal variations of water vapor have not been investigated in the past due to the lack of observation is not correct.

2. L12-13: Is this a new finding?

3. L13: high (surface) pressure?

4. L26: The statement that water vapor in the free troposphere contributes to the water vapor feedback through latent heat processes is confusing, because the water vapor feedback is associated with radiative processes. Please clarify.

5. L30-34: These sentences describe the water vapor feedback. What is the difference from the lines 24-28? The first paragraph of introduction needs to be reorganized.

6. L40: Do the authors mean Kottayil et al. (2013)?

7. L39-44: The transition is not logical. It seems that sentences are missing between the two sentences.

8. L46: Kottayil et al. (2013)?

9. L67: the lack of "adequate" observations?

10. L84: six instead several?

11. L89: Does "upper" mean channel 1?

12. L89-90: Please consider adding additional y-axis (altitude) to the right in Figure 1.

13. L105: Does "i" denote the earth incidence angle?

14. L115-116: However, the radiative transfer calculations are used to derive the empirical coefficients in Eq. 1 and to determine the thresholds for excluding surface affected radiances. Therefore, the phrases "to avoid any possible errors due to the radiative transfer calculations" need to be changed.

15. L124: Do "upper" and "lower" here have the same meaning as in the line 89?

16. L158: Tian et al. (2004)

17. L173-174: Please specify the channels.

18. L197-198: redundant (lines 194-195)

19. L205: high (surface) pressures?

20. L212: water vapor or moisture instead of humidity?

21. L228: Figure 8

22. L229: Figure 9

23. L236-237: redundant (lines 234-236)

24. L238-242: The errors are not significant because the accuracy of SAPHIR measurements is roughly 5% in RH space (line 73). In that case, polar-orbiting satellite observations are sufficient to determine the daily mean. Please discuss in the paper.

25. L292: Consider replacing South East Asia by the maritime continent (also in Table 2).

26. L317: Figure 13 does not indicate the early morning maximum/afternoon minimum of RH over the South Atlantic (cf. line 320).

27. L319: Figure 13 shows the afternoon minimum of RH over Amazon and South East Asia.

28. L334-335: Please clarify.

29. L345: The distribution is different between the South Atlantic (right-skewed?) and South East Asia (left-skewed?) in Figure 14. Please clarify and reorganize.

30. L368: Do the authors argue that microwave radiances are significantly affected by thin clouds? If so, what is the advantage of microwave measurements over the infrared observations?

31. L378-383: It is difficult to figure out how the diurnal variation of tropospheric humidity is related to global warming. Please discuss in the manuscript.

32. L395: high (surface) pressure?

33. L396: Please clarify "significant" regions.

34. L488: Coauthors are missing.

35. Figure 2: Please correct typo (y-axis).

36. Figure 3: Are the histograms independent of latitudes and seasons?

37. Figure 4: The range of simulated Tb is 240-280 K in Figure 3. In contrast, the range is narrower here. Why are they different?

38. Figures 5-12: Please include tick labels for longitude and latitude in Figures 5-12.

39. Figures 6-7: Please specify the time period for the mean daily relative humidity.

40. Figure 13: It seems that the diurnal variation is only evident over regions of high elevation.

---

## Author Comment (AC3) · 11 Apr 2016

**3 Reviewer #3**

We truly appreciate the reviewer for carefully reading the entire manuscript and making very constructive recommendations to improve the study as well as the manuscript. We have adopted all the recommendations in the revised version and have also provided point-to-point responses in the following.

This study uses microwave measurements from SAPHIR onboard the Megha- Tropiques satellite to investigate the diurnal variation of water vapor in the tropics. In this study, the limb effect-corrected observed radiances were transformed into layer- averaged relative humidity and then partitioned into 24 bins of local observation time. The authors determined the phase and amplitude of diurnal variation by fitting the Fourier series to the binned data, and showed a large inhomogeneity in the diurnal variation of tropospheric relative humidity in the tropics. Although there are some is- sues that the authors need to clarify, the results presented in this study appear to help improve our understanding of the diurnal variation of water vapor.

Many thanks for the positive feedback!

1. Motivation of the study: Although the authors argues that the diurnal variation of water vapor has not been investigated due to the lack of observations, the argument is not true given the previous studies addressed in the manuscript. Also, the diurnal variation presented in the manuscript is very weak over most regions of the tropics, raising a question on the need of investigation on the diurnal variation of water vapor. Moreover, it is unclear in what ways the diurnal variation of water vapor is important.

We appreciate the reviewer for this valid comment. We have revised the introduction to better clarify the novelty of the current study. We have also more properly cited the previous publications (mainly conducted using the IR measurements) in the revised version. However, since these changes are major and are introduced in several different places, we have directly incorporated them in the revised version and have not copied the revised text here.

2. The authors argue that their results are superior to previous studies based on IR measurements or multi-instrument microwave measurements. However, it doesn't seem that the differences from the previous studies are discussed comprehensively in the manuscript. Therefore, it is not certain whether this study advances our understanding of the diurnal variation of water vapor

in the tropics.

In the revised version, in addition to citing previous publications, we have also better compared our results with the previous studies.

3. The manuscript documents the peak time and amplitude of the diurnal variation in the tropics, but does not provide reasonable physical mechanisms responsible for the spatial and altitude discrepancies in the diurnal variation. Also, it is unclear why some parts of the tropics have an early morning maximum of the tropospheric relative humidity.

We have now provided more explanation on the physical mechanisms behind the diurnal cycle of RH. We have also provided explanation for the early morning peak time which is consistent with previous studies and is correlated with the early morning peak time of convergence zones (e.g., Haffke et al 2015).

4. Given the seasonal migration of ITCZ, the amplitude and peak time of the diurnal variation for a given season might be different from that for the annual mean. However, this aspect is not investigated in the manuscript.

We agree with the reviewer regarding the impact of ITCZ on the diurnal cycle of RH, therefore in addition to the mean annual values, we have also included the diurnal amplitude and peak time for the months of December/January as well as June/July in the revised version. As the reviewer pointed out, the results change with the season due to shift in ITCZ.

**Specific points:**

1. L1-2: The argument that the diurnal variations of water vapor have not been investigated in the past due to the lack of observation is not correct.

As we mentioned in response to the reviewer's general comments, we rewrote the introduction to better clarify the new aspects of our study.

2. L12-13: Is this a new finding?

We slightly changed the sentence to emphasize that this is not a new finding:

> The results showed that the wet regions are normally associated with convective regions, and the dry regions with the subsidence regions which are consistent with the previous studies.

3. L13: high (surface) pressure?

Changed to "the subsidence regions"

4. L26: The statement that water vapor in the free troposphere contributes to the water vapor feedback through latent heat processes is confusing, because the water vapor feedback is associated with radiative processes. Please clarify.

We changed it to "radiative processes" and also rewrote the introduction for a better clarification.

5. L30-34: These sentences describe the water vapor feedback. What is the difference from the lines 24-28? The first paragraph of introduction needs to be reorganized.

We have revised and reorganized the introduction as suggested by the reviewer.

6. L40: Do the authors mean Kottayil et al. (2013)?

Many thanks to the reviewer for noting the error. It is now corrected.

7. L39-44: The transition is not logical. It seems that sentences are missing between the two sentences.

We added a sentence to provide a logical transition as follows:

> One exception is Kottayil et al. (2013) that used multi-instrument microwave measurements from five polar-orbiting satellites to investigate the diurnal variation of brightness temperature (Tb) over the globe. However, other issues are involved when data from polar-orbiting instruments are utilized. First, polar-orbiting satellites only overpass each location twice a day, thus even a constellation of five satellites do not properly represent the diurnal variation of RH (e.g. see Figure 1 in Kottayil et al. (2013) for the temporal coverage in different years).

8. L46: Kottayil et al. (2013)?

Changed the citation to Kottayil et al. (2013)

9. L67: the lack of ''adequate'' observations?

We amended the sentence and now it reads "lack of adequate observations.

10. L84: six instead several?

Done!

11. L89: Does ''upper'' mean channel 1?

11

Yes! The text was amended as follows to reflect this:

> Figure 1 shows the weighting functions for the SAPHIR channels which are roughly sensitive to upper (channel 1 peaking around 10 km) to lower troposphere (channel 6 peaking around 2 km).

**12. L89-90: Please consider adding additional y-axis (altitude) to the right in Figure 1.** Done!

**13. L105: Does ''i'' denote the earth incidence angle?**
The index "i" was unnecessary. We have removed it in the revised version.

**14. L115-116: However, the radiative transfer calculations are used to derive the empirical coefficients in Eq. 1 and to determine the thresholds for excluding surface affected radiances. Therefore, the phrases ''to avoid any possible errors due to the radiative transfer calculations'' need to be changed.**
We amended that as

> Since the SAPHIR data do not suffer from scan asymmetry, we preferably used the satellite data to develop the limb-correction technique.

**15. L124: Do ''upper'' and ''lower'' here have the same meaning as in the line 89?**
The upper and lower had been mistakenly switched in L124. Now it reads

> ... screen-out the clouds using the differences between Tb's of an upper channel (Tb2, channel 2 operating at 183±1.10 GHz) and a lower channel (Tb5, channel 5 operating at 183±6.8 GHz).

**16. L158: Tian et al. (2004)**
Done

**17. L173-174: Please specify the channels.**
Done

**18. L197-198: redundant (lines 194-195)**
Removed the redundancy. Now it reads as

> As shown, on average, 100 to 300 observations are retained for each bin per hour.

19. L205: high (surface) pressures?
Changed to "subsidence regions"

20. L212: water vapor or moisture instead of humidity?
Changed to water vapor.

21. L228: Figure 8 22. L229: Figure 9

Many thanks to the reviewer for noting the errors. Both errors are fixed.

23. L236-237: redundant (lines 234-236)
Removed 236-237.

24. L238-242: The errors are not significant because the accuracy of SAPHIR measurements is roughly 5% in RH space (line 73). In that case, polar-orbiting satellite observations are sufficient to determine the daily mean. Please discuss in the paper.
Yes, we also believe that polar-orbiting satellites may be used to derive the mean tropospheric humidity to some extent but not the amplitude and peak time. We also need to mention that there is probably a small systematic error in the SAPHIR data but when looking at the peak and amplitude the systematic error is not important because it is canceled out when we take the differences. The random error is also mostly canceled out so the actual error is really very small and negligible. Therefore, we added the following statement to clarify this:

> These results show that measurements from polar-orbiting satellites may be used to derive the mean tropospheric humidity in the tropical region to some extent. However, polar-orbiting satellites may not provide a good picture of peak time and amplitude, because these parameters show a large spatial inhomogeneity and obviously depend on the satellite overpass time.

25. L292: Consider replacing South East Asia by the maritime continent (also in Table 2).
Done

26. L317: Figure 13 does not indicate the early morning maximum/afternoon minimum of RH over the South Atlantic (cf. line 320).
Many thanks to the reviewer for carefully checking the results. We realized that it is difficult to directly interpret diurnal variations presented in Figure 13. We replotted the RH values by either removing the mean RH

13

from the diurnal cycle or scaling the RH values between a minimum and maximum. It turned out that the latter, Figure 2, better indicates the relative diurnal cycle of RH over all regions. Figure 2 is also included in the revised version of the manuscript. We agree with the comment and accordingly the text has been revised based on results presented in both Figure 13 and the new figure to better explain the diurnal variation of RH.

27. **L319: Figure 13 shows the afternoon minimum of RH over Amazon and South East Asia.**

We agree with the reviewer. As mentioned, we are providing a new figure, Figure 2, that better shows the diurnal cycle of RH over all the regions.

28. **L334-335: Please clarify.**

We amended the sentence as follows for a better clarification:

> As shown the distributions of RH with respect to ice and water are similar in the lower troposphere (channels 5 and 6). This is because the relative humidity values calculated using Equation 1 are nearly the same over ice and liquid for the expected range of brightness temperatures in lower troposphere. This is an indication that the transition from ice to liquid is performed smoothly.

29. **L345: The distribution is different between the South Atlantic (right-skewed?) and South East Asia (left-skewed?) in Figure 14. Please clarify and reorganize.**

We thank the reviewer for noting the error. We amended it as follows:

> The distribution is left-skewed (the left tail is longer) for Amazon and the Maritime Continent, and right-skewed over the rest of the selected regions.

30. **L368: Do the authors argue that microwave radiances are significantly affected by thin clouds? If so, what is the advantage of microwave measurements over the infrared observations?**

We meant "thin" from a microwave instrument perspective which is still generally a thick cloud. We removed the word thin to avoid any confusion.

31. **L378-383: It is difficult to figure out how the diurnal variation of tropospheric humidity is related to global warming. Please discuss in the manuscript.**

We completely rewrote that paragraph to better explain the relation between the observations and climate simulations.

[Figure]

Figure 2: Diurnal variation of RH scaled between -100 and 100 individually for each region.

32. L395: high (surface) pressure?
We changed that to subsidence regions.

33. L396: Please clarify ''significant'' regions.
We changed it to several regions.

34. L488: Coauthors are missing.
Coauthors are now included.

35. Figure 2: Please correct typo (y-axis).
Done!

36. Figure 3: Are the histograms independent of latitudes and seasons?
Yes, they are independent of the season and latitude. We have used a large subset of data covering different atmospheric conditions.

37. Figure 4: The range of simulated Tb is 240-280 K in Figure 3. In contrast, the range is narrower here. Why are they different?
In Figure 3, only a small percentage of the data are between 270-280K, which are not shown in Figure 4 because Figure 4 is a density plot.

38. Figures 5-12: Please include tick labels for longitude and latitude in Figures 5-12.
Done!

39. Figures 6-7: Please specify the time period for the mean daily relative humidity.
We have used the data for the period January 2012 to September 2015. This is now clarified in the captions as well as in the text in Section Satellite Data.

40. Figure 13: It seems that the diurnal variation is only evident over regions of high elevation.
The diurnal variation is dominant over high elevations but also the tropical lands such as Africa. However, the diurnal variation, though exist, is very weak over oceans.

---

## Author Comment (AC2)

**2 Reviewer #2**

We greatly appreciate the reviewer's effort to review the manuscript and make recommendations to improve the paper. The reviewer's points were well taken and we have adopted all the reviewer's comments in the revised version. The main concern of the reviewer is that we have not properly cited the previous publications. We agree that we should have either cited the previous publications conducted using infrared measurements or should have provided an explicit justifications for not doing so. Nevertheless, we are providing very comprehensive answers to the comments, hoping to fully address the reviewer's concerns. In order to adopt the reviewer's comments, we have particularly revised the introduction to better explain the novelty of our work. In addition to citing several new relevant papers in the introduction and elsewhere, we have also considered analyses and results provided in the previous publications for the interpretation of our results.

General comments: The paper addresses an interesting topic, it attempts to analyse the diurnal cycle of humidity in tropical regions in a more coherent manner with a multi-channel microwave instrument in a drifting orbit. I agree with the authors that there is still scope for better analyses of the diurnal cycle of tropospheric humidity, and the mesurements used in this paper provide a good basis for such an analysis.

We thank the reviewer for the encouragement.

The authors state: Despite the importance of water vapor especially in the tropical region, the diurnal variations of water vapor have not been investigated in the past due to the lack of observations. - This not true. - Let me give you a few examples of papers that analyse the diurnal cycle of upper tropospheric moisture, the first one goes back more than 20 years.

We agree that there has been several previous efforts to evaluate the diurnal cycle of relative humidity mainly using infrared measurements. We have now cited these publications in the introduction and have also benefited from the discussions provided in some of these publications that try to connect the RH distribution with the physics and dynamics of the atmosphere. In addition to citing publications introduced by the reviewer, we have also included several recent publications to diversify our references. That being said, we would like to emphasize that for the reasons explained below, the IR and MW measurements are expected to yield different results, therefore the disagreement between the results should be considered as the instrumental differences.

- the cloud screening removes a large portion of the IR measurements

especially over convective regions. The rejected observations normally represent moist conditions, therefore the IR results only represent dry conditions. For instance, John et al. (2011) indicated that the IR cloud screening introduces on average around 10% systematic error in the upper tropospheric humidity values. It is clearly shown in John et al. (2011) that the cloud screening especially removes most of the data over the convective regions causing a large systematic bias in the RH analysis for the convective regions. It should be noted that among the channels, the upper tropospheric channels are less sensitive to clouds than the lower channels, because the weighting functions for the upper channels normally peak above the clouds, therefore it is expected that the dry bias due to cloud screening is even larger for the middle and lower tropospheric channels.

- it should be noted that the cloud screening not only impacts the RH amplitude by removing the moist conditions, but also impacts the diurnal peak time. For instance, Figure 1 shows an example of the impact of applying different cloud thresholds on the diurnal cycle of relative humidity. As shown the diurnal peak time significantly changes with the threshold used to filter the clouds.

[Figure]

Figure 1: Impact of cloud screening on the diurnal peak time of upper tropospheric humidity. Image contributed by Ajil Kottayil, Cochin University of Science and Technology, India

My second major concern is that the analysis is too descriptive with little attempts to explain the observations in terms of meteorology/physics.

We have now provided more information and explanation for the physical mechanisms behind the results. Besides, we have also provided details regarding how the results are connected with mesoscale features of the atmosphere. However, we need to emphasize that the goal of the current study was to provide an observational analysis as well as a clear picture of the diurnal cycle of RH in the tropical region using unbiased observations. We agree with the reviewer that there is still a need to conduct work on the impact of meteorology/physics on the diurnal cycle of RH. However, this cannot be accomplished using the data provide by SAPHIR instrument nor was the goal of the current study. This would require combining data from models/reanalyses and measurements but as it has been shown in previous publications (e.g., Chung et al. 2013), the diurnal cycle in most models/reanalyses is not expected to be accurate.

---

## Author Response (AR2)

**Reviewer #3**

We sincerely appreciate the reviewer for carefully reading the manuscript and providing feedback. We have considered all the comments in the revised manuscript. A short answer to the comments is also given below.

L14-17: Please specify the new findings in this study. Although the authors reported a large spatial inhomogeneity in the characteristics of relative humidity diurnal cycle over the tropics, previous studies have also documented the presence of the spatial inhomogeneity. In addition, physical mechanisms responsible for the large spatial inhomogeneity need to be presented.

We have substantially revised the abstract to better explain the results and the mechanism behind the results.

L481-485: It is unclear in what ways the diurnal variation of water vapor plays an important role in global warming and climate change predictions.

We changed that paragraph as follows:

> Water vapor significantly influences the Earth's climate because of its greenhouse effect. Water vapor is also important to the global water and energy budget. Free tropospheric water vapor, especially in the tropical region, significantly contributes to the water vapor feedback through radiative processes (Trenberth et al., 2009; Dessler et al., 2008). Diurnal cycle of humidity influences diurnal variations in other geophysical variables such as precipitation, and convective activities (Chung et al. 2007). Despite the importance of water vapor and its diurnal variation, current climate and numerical weather prediction models do not adequately simulate the diurnal variation of tropospheric humidity (Chung et al. 2007). As the simulations of climate models are improved, more accurate observations are required to verify the validity of simulations.

Please highlight new findings in the section of Conclusions and Summary. Consider discussing the implication of new findings for global warming and climate change predictions. In addition, the authors need to discuss physical mechanisms responsible for the large spatial inhomogeneity in the diurnal cycle of relative humidity over the tropics.

We did a major revision in the conclusion section to include the new findings and also to discuss the physical mechanisms behind the results.

*Figure 2: Please indicates what EIA stands for in the figure caption.*
Done!

*Please correct typos.*
We carefully read the entire manuscripts and have corrected the typos.

*Please check whether cited literature is relevant because in some cases seemingly irrelevant literature was cited in the manuscript.*

We carefully checked the manuscript (especially the introduction) and have removed several of the references that were not directly relevant to the text.

[revised manuscript text omitted]